# Two-step chromosome segregation in the stalked budding bacterium *Hyphomonas neptunium*

Alexandra Jung[1,2], Anne Raßbach[1,2], Revathi L. Pulpetta[1,2], Muriel C.F. van Teeseling[1,2], Kristina Heinrich [1,2], Patrick Sobetzko[1,3], Javier Serrania[1,3], Anke Becker[1,3] & Martin Thanbichler [1,2,3]

Chromosome segregation typically occurs after replication has finished in eukaryotes but during replication in bacteria. Here, we show that the alphaproteobacterium *Hyphomonas neptunium*, which proliferates by bud formation at the tip of a stalk-like cellular extension, segregates its chromosomes in a unique two-step process. First, the two sister origin regions are targeted to opposite poles of the mother cell, driven by the ParAB*S* partitioning system. Subsequently, once the bulk of chromosomal DNA has been replicated and the bud exceeds a certain threshold size, the cell initiates a second segregation step during which it transfers the stalk-proximal origin region through the stalk into the nascent bud compartment. Thus, while chromosome replication and segregation usually proceed concurrently in bacteria, the two processes are largely uncoupled in *H. neptunium*, reminiscent of eukaryotic mitosis. These results indicate that stalked budding bacteria have evolved specific mechanisms to adjust chromosome segregation to their unusual life cycle.

[1] Faculty of Biology, Philipps University, Karl-von-Frisch-Straße 8, 35043 Marburg, Germany. [2] Max Planck Institute for Terrestrial Microbiology, Karl-von-Frisch-Straße 10, 35043 Marburg, Germany. [3] Center for Synthetic Microbiology, Hans-Meerwein-Straße 6, 35043 Marburg, Germany. Correspondence and requests for materials should be addressed to M.T. (email: thanbichler@uni-marburg.de)

One of the biggest challenges that cells face is to accommodate their chromosomal DNA and to ensure its faithful replication and segregation during cell division. Eukaryotic cells cope with this issue by temporally uncoupling the replication and segregation process. After they have reached the end of S-phase, their sister chromatids cohere for an extended period of time, condensing into compact structures that are finally pulled apart simultaneously to opposite cell halves by the mitotic spindle apparatus[1,2]. In bacteria, by contrast, sister loci typically experience no or only a very brief period of cohesion and segregate almost immediately to their final destination in the incipient daughter cell compartments[3–5]. Bacterial chromosome replication and segregation thus occur concurrently, and various mechanisms are in place to closely coordinate these processes in time and space.

Most bacteria possess one circular chromosome that is arranged into a ring-like superstructure whose overall organization roughly reflects the circular chromosomal map[3,6–8]. Its duplication initiates at a single origin of replication (*ori*) and then proceeds bidirectionally along the two chromosomal arms until the replication forks meet in the terminus (*ter*) region. Although the subcellular location of these landmark sites varies among species[3,6], the majority of organisms analyzed to date show a longitudinal (*ori*–*ter*) pattern, in which the *ori* and *ter* regions are located at or close to opposite cell poles, while the two chromosomal arms are arranged side by side in-between these two fixed points[4,9–12]. After replication initiation, one of the duplicated *ori* regions traverses the cell towards the opposite end. The remaining parts of the chromosome then follow successively as replication proceeds, thereby gradually displacing the *ter* region towards midcell and re-establishing the original *ori*–*ter* pattern in the two daughter cells[3,8]. Alternatively, bacteria can display a transverse (left-*ori*-right) chromosome arrangement, with the *ori* and *ter* regions positioned around midcell and the two chromosomal arms segregated to opposite cell halves[13–16]. Some species switch between these patterns dependent on their cell cycle or developmental state[17–21].

The mechanisms underlying bacterial chromosome segregation are still incompletely understood and appear to vary between different lineages. In many species, *ori* segregation is driven by the ParABS system[3,6] and/or the condensin-like SMC complex[6,22]. Various factors, such as entropic forces, transcription, and DNA condensation may then act together to achieve bulk chromosome segregation[23–25], supported by the activity of DNA topoisomerases, which facilitate the resolution of tangled DNA regions[26]. Finally, after decatenation and chromosome dimer resolution[7], the *ter* regions are partitioned with the help of DNA translocases that help to clear the division site of non-segregated DNA[27,28].

ParABS partitioning systems consist of three components: (i) multiple copies of a centromere-like sequence motif (*parS*) that are distributed within the *ori* region[29–31], (ii) a DNA-binding protein (ParB) that binds specifically to these *parS* sites and then further spreads into the adjacent regions of the nucleoid[17,29,30,32,33], and (iii) a P-loop ATPase (ParA) that acts as a molecular switch mediating the partitioning process[34–37]. During origin segregation, ParA dimers bind non-specifically to the nucleoid, forming a concentration gradient with a maximum at the new cell pole and a minimum at the moving *ori* region[37]. In addition, they interact with the *ori*-associated ParB·*parS* complex and tether it to the nucleoid surface. ParB, in turn, stimulates the ATPase activity of adjacent ParA dimers, leading to their disassembly. As a consequence, the ParB·*parS* complex is released and free to interact with ParA dimers in its vicinity. Iteration of this cycle is thought to promote the directed, ratchet-like movement of the segregating *ori* region along the ParA dimer gradient[34–36,38–40]. In many species, the segregation process is supported by polar landmark proteins that sequester the ParB·*parS* complex at the cell poles[41–46], as exemplified by the polymeric scaffolding protein PopZ from the alphaproteobacterial model organism *Caulobacter crescentus*[43,44]. Interestingly, apart from anchoring the chromosomal origin regions, PopZ also helps to capture monomeric ParA molecules. It thus prevents the reassembly of ParA-ATP dimers in the wake of the moving ParB·*parS* complex, thereby ensuring the directionality of the segregation process[35,36,47].

Up to this point, bacterial chromosome organization and dynamics have been mainly studied in rod-shaped model organisms that divide by binary fission[6]. However, many species have more complex morphologies and life cycles. A prominent example is the marine bacterium *Hyphomonas neptunium*, a relative of *C. crescentus* that proliferates by an unusual budding mechanism in which new offspring emerges from the tip of a stalk-like cellular extension[48–50]. Cell division at the bud neck generates a flagellated, mobile swarmer cell and an immobile stalked cell. Whereas the stalked cell immediately enters the next reproductive cycle, the swarmer cell first needs to shed its flagellum and form a new stalk before it can initiate bud formation[49,51]. The mechanisms that transfer large cellular components such as chromosomal DNA from the mother cell to the nascent bud compartment are still unknown. However, the recent establishment of a genetic system for *H. neptunium*[52] now offers the possibility to dissect the cell biology of this species at the molecular level.

In this study, we aim to unravel how the organization and dynamics of chromosomal DNA have adapted to the specific needs of the *H. neptunium* life cycle. We demonstrate that chromosome segregation in *H. neptunium* occurs in a unique two-step process. Swarmer cells initially contain a single chromosome that shows a circular arrangement in the cell, with its *ori* region positioned in the vicinity of the old cell pole. DNA replication initiates shortly after the onset of stalk formation. The two sister *ori* regions are then first segregated within the mother cell, in a manner dependent on the ParABS system. During this process, one of the ParB·*parS* complexes is moved to the stalked mother cell pole, where it remains fixed for an extended period of time. Later in the cell cycle, it then regains mobility and rapidly moves through the stalk into the nascent bud compartment, driven by an unknown mechanism. Importantly, this second segregation step initiates close to the end of S-phase, indicating that the partitioning of sister chromosomes to the incipient daughter cell compartments is largely uncoupled from DNA synthesis.

## Results

**H. neptunium ParB binds to *parS* sites in vitro.** To gain first insight into the organization of the *H. neptunium* chromosome, we aimed to determine the position of its replication origin. Bioinformatic analyses based on established *ori* markers such as the GC skew[53,54], the gene of the replication initiator DNA, and the position of DnaA binding sites[53,55,56] did not yield a clear result. Therefore, we used marker frequency analysis to experimentally pinpoint the chromosomal *ori* region, based on the idea that *ori*-proximal DNA is duplicated earlier than *ter*-proximal DNA and thus, on average, more abundant in the population. For this purpose, chromosomal DNA was prepared from synchronized *H. neptunium* cells shortly after their entry into S-phase and subjected to high-throughput sequencing. Analysis of the read frequencies yielded a bell-shaped curve with a maximum around 0° and close to *hemE*, a gene flanking *ori* in *C. crescentus* (Fig. 1a), indicating that replication initiates at similar sites in these two species.

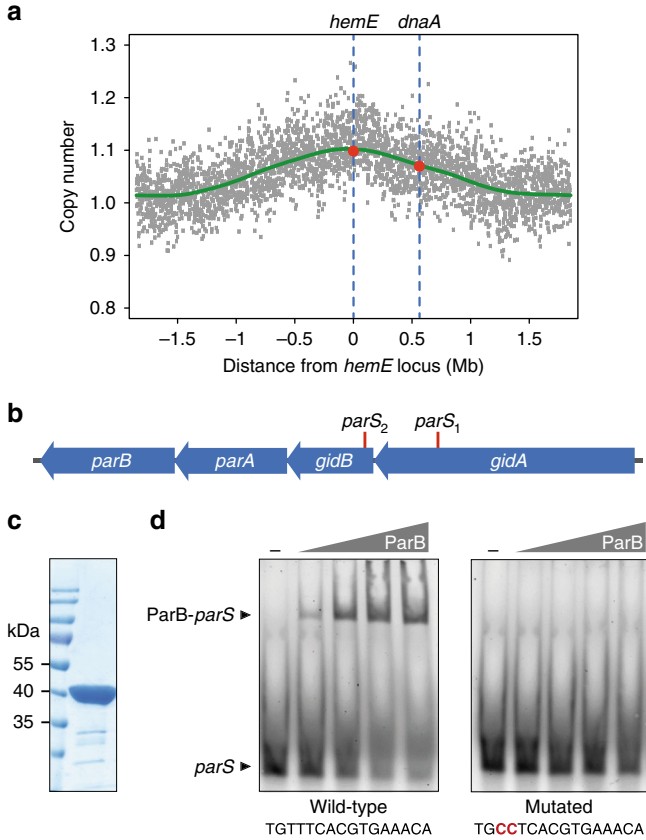

**Fig. 1** Characterization of the chromosomal *ori* region of *H. neptunium*. **a** Location of the chromosomal replication origin. The plot shows the relative abundance of chromosomal loci in growing *H. neptunium* wild-type cells, as determined by sequencing-based marker frequency analysis. The green line indicates the average of the data points. The values obtained for the *hemE* and *dnaA* loci are indicated by red dots. **b** Schematic representation of the *parAB* locus. The positions of the two *parS* sites identified in the *H. neptunium* genome sequence are indicated. **c** Purity of the ParB preparation used in the binding assay. Purified ParB-His$_6$ (10 μg) was subjected to SDS-PAGE and stained with Coomassie blue. A molecular weight standard is shown for comparison. **d** Interaction of ParB with its *parS* target sequence. Cy3-labeled double-stranded DNA oligonucleotides (10 nM) carrying the wild-type (left) or a mutated (right) *parS* sequence, respectively, were incubated with increasing concentrations of ParB-His$_6$ (0, 0.05, 0.1, 0.3, and 0.6 μM). Nucleoprotein complexes were separated from the free oligonucleotides by PAGE and visualized by fluorescence imaging. Red letters indicate substitutions in the *parS* sequence

**ParB·parS complexes are segregated in a two-step process.** To clarify the contribution of ParB to chromosome segregation in

Searching for a potential chromosome segregation system, we identified a *parAB* operon in close proximity (~7 kb) of the *hemE* gene. Moreover, we located two potential *parS* sites upstream of this operon, which perfectly corresponded to the previously published global consensus sequence[57] (Fig. 1b). To verify the functionality of the *H. neptunium* ParABS system, we tested whether purified ParB (Fig. 1c) was able to interact with these predicted *parS* sites in vitro using an electrophoretic mobility shift assay. The protein indeed bound to a double-stranded DNA oligonucleotide containing the wild-type *parS* sequence (Fig. 1d). This interaction was abolished by mutation of two highly conserved bases in the 5′ region of the motif, demonstrating that ParB specifically associates with *parS* in *H. neptunium*.

*H. neptunium*, we generated a strain producing a functional ParB-YFP fusion in place of the native ParB protein. Fluorescence imaging revealed that the fusion protein formed well-defined foci, indicating its assembly into nucleoprotein complexes at the origin-associated *parS* sites. Swarmer cells showed a single ParB·*parS* complex that was positioned close to the old cell pole (Fig. 2a). At the onset of stalk formation, and concomitant with the entry into S-phase (see below), this complex was duplicated. One of the copies moved across the cell toward the stalked pole, where it remained until the cell had finished stalk formation and formed a visible bud. At this point, a second phase of DNA segregation initiated, in which the moving ParB·*parS* complex was transported through the stalk and finally attached to the old (stalk-distal) pole of the nascent bud compartment. This localization pattern was corroborated by quantitative analysis of the ParB-YFP signals in a random population of cells (Fig. 2b). *H. neptunium* is thus a monoploid bacterium (see also Supplementary Fig. 1) that initiates chromosome replication only once per division cycle and then segregates its two sister chromosomes in a unique two-step process.

To better resolve the dynamics of origin segregation, we followed the movement of ParB-YFP foci by time-lapse analysis using shorter (5 min) time intervals. Tracking of the fluorescence signals revealed that it took 10–55 min to fully segregate newly synthesized sister *ori* regions within the mother cell, with an average segregation time of $30 \pm 13$ min ($n = 19$; ±SD) and a speed of $0.05 \pm 0.03$ μm min$^{-1}$ ($n = 17$). The focus positioned at the stalked mother cell pole then remained stationary for $58 \pm 24$ min ($n = 19$), before it started to enter the stalk and migrate towards the nascent bud compartment. Unlike the first segregation step, the translocation of the *ori* region through the stalk and its sequestration to the old bud cell pole were rapid processes that lasted only a few minutes in total (Fig. 2c). Assuming that the two replication forks progress at rates of ~400 bp s$^{-1}$ [10,58–60], the 3.7 Mb chromosome of *H. neptunium* should be duplicated within approximately 80 min. Thus, the second segregation step likely initiates around or after the end of S-phase. Measurements of the cellular dimensions indicated that the transition to the second segregation step occurred once the bud had reached ~55% of the mother cell width (Fig. 2d), ensuring that the bud is large enough to accommodate the chromosomal DNA.

It was difficult to monitor the second segregation step in cells grown in rich medium, because stalks were relatively short under these conditions. To solve this issue, we made use of the fact that stalks elongate considerably upon phosphate limitation. When phosphate-starved cells were transferred back to nutrient-replete conditions to promote cell cycle progression, ParB-YFP remained stalled at the stalk base until the bud reached its critical size. Subsequently, it traversed the stalk in a rapid and directed manner, reaching speeds of ~1 μm min$^{-1}$ (Fig. 2e). These findings strongly suggest that origin translocation through the stalk is driven by an active mechanism.

**ParA is essential for origin segregation in the mother cell.** To further investigate the role of ParA and ParB, we aimed to generate mutants carrying in-frame deletions in the corresponding genes. However, all our attempts were unsuccessful, suggesting that the ParABS system is essential for viability in *H. neptunium*. We therefore set out to construct conditional mutants that expressed a *parA-venus* or *parB-yfp* fusion, respectively, under the control of a zinc-inducible promoter to compensate for the deletion of the native gene. In the case of *parB*, this approach failed, even though the fusion construct was fully functional when expressed from the native locus (see Fig. 2), suggesting that the proper level and timing of expression are critical for ParB

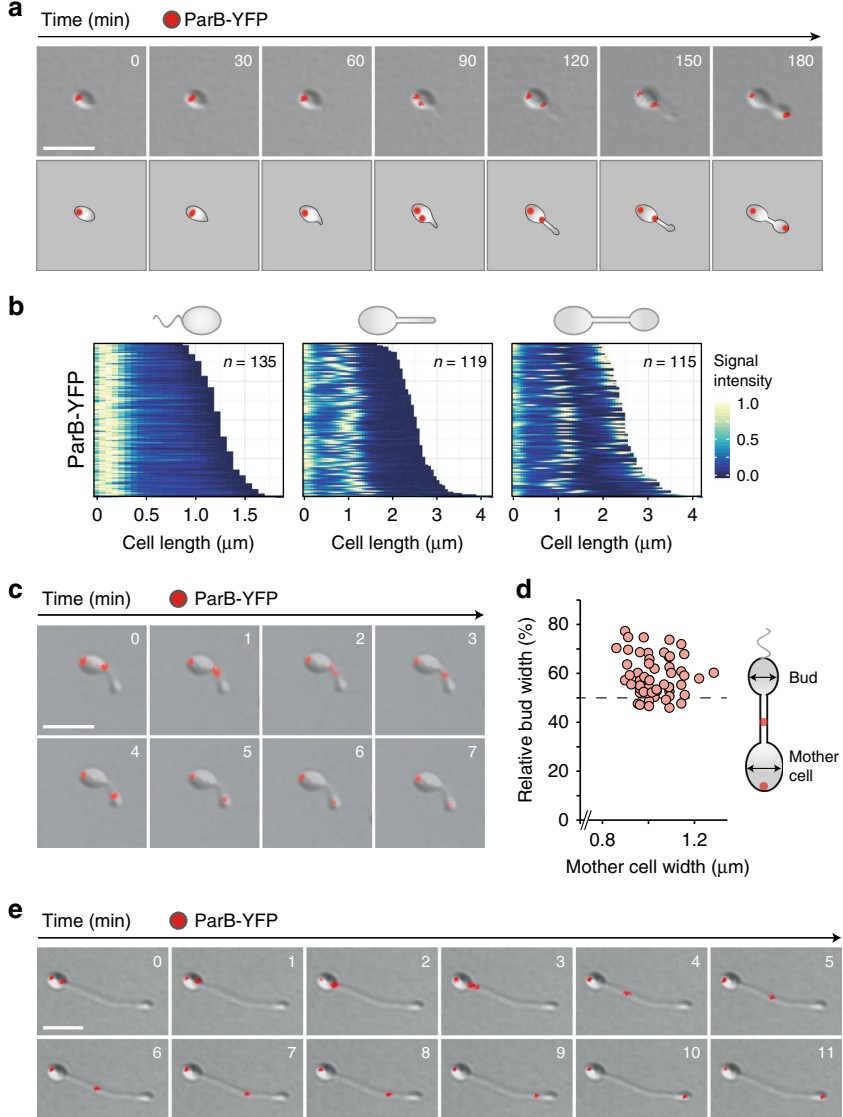

**Fig. 2** Two-step segregation of the sister *ori* regions. **a** Time-lapse series showing the localization of ParB-YFP over the course of the cell cycle. *H. neptunium* KH22 (*parB-yfp*) was grown in MB medium, transferred to an MB-agarose pad, and imaged at 30 min intervals. Shown are overlays of differential interference contrast (DIC) and fluorescence micrographs as well as schematics of the fluorescence patterns observed. Bar: 3 μm. **b** Demographic analysis of ParB-YFP localization in swarmer (left), stalked (middle), and budding (right) cells of strain KH22. The fluorescence intensity profiles obtained for each cell type were sorted according to cell length and stacked on each other, with the shortest cell shown at the top and the longest cell shown at the bottom. Note that due to the variable length of the stalk, the total cell length only roughly corresponds to the developmental state of the cell. **c** Time-lapse series showing the movement of ParB-YFP through the stalk of strain KH22. Cells were imaged at 1 min intervals. Shown are overlays of DIC and fluorescence images. Bar: 3 μm. **d** Relative size of the bud at the onset of the second segregation step. Strain KH22 was imaged at 5 min intervals. At the first time point at which ParB-YFP had left the stalk base to migrate toward the bud, the widths of the mother cell and bud compartments were measured. The graphs show the relative width of the bud plotted versus the absolute width of the mother cell ($n = 57$). The average of the data is 60 ± 7%, corresponding to a total width of 0.61 ± 0.08 μm. **e** Time-lapse series following *ori* translocation through the elongated stalk of a phosphate-starved cell. Strain KH22 was grown in minimal medium lacking phosphate for 22 h. Subsequently, MB medium was added to a final concentration of 20% and growth was continued for 40 min. After transfer of the cells onto an MB-agarose pad, images were taken at 1 min intervals. Shown are overlays of DIC and fluorescence micrographs. Bar: 3 μm

function. By contrast, we readily obtained a strain carrying a conditional *parA-venus* allele (Fig. 3a). Under inducing conditions, the cells largely showed wild-type morphology and growth rates, indicating that ParA was not affected by addition of the fluorescent tag (Fig. 4a–c).

The availability of a functional fluorescent protein fusion offered the possibility to investigate the dynamics of ParA localization over the course of the cell cycle. Time-lapse and

demographic analyses revealed that the majority of swarmer cells and early stalked cells showed a bright focus at the flagellated pole, accompanied by fainter fluorescent patches in other regions of the cell (Fig. 3b, c). During later stages of stalk formation, the intensity of the polar signal decreased, whereas the number and intensity of non-polar patches increased. At the very beginning of the budding process, the majority of ParA-Venus molecules relocated to the nascent bud. Later on, as the bud increased in

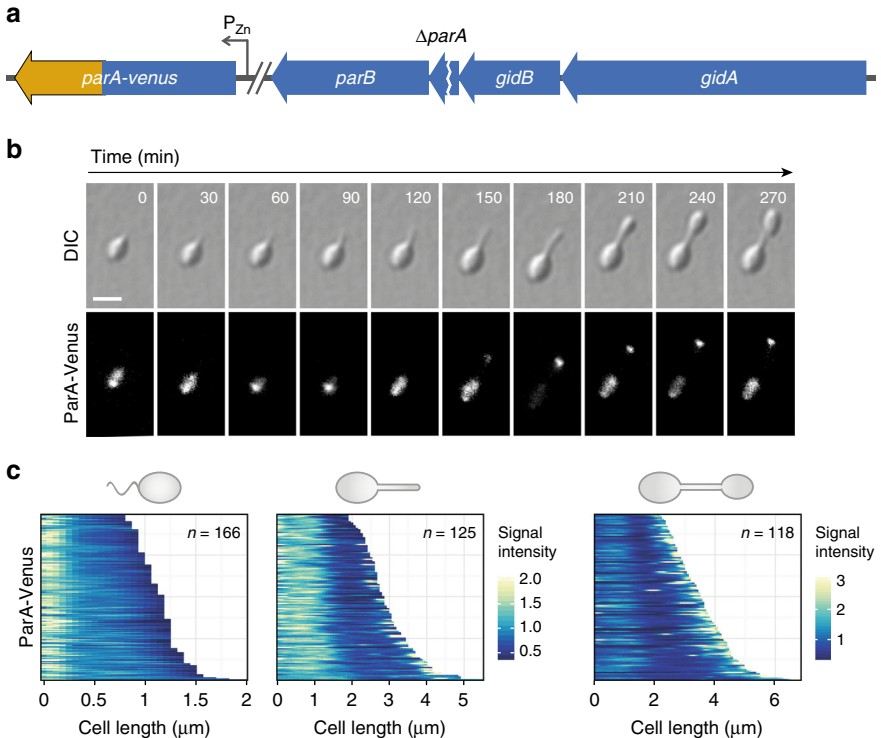

**Fig. 3** Localization dynamics of ParA. **a** Schematic explaining the construction of the strain used to follow the localization of ParA (AJ46). **b** Time-lapse series showing the localization of ParA-Venus at different stages of the cell cycle. Cells of strain AJ46 ($\Delta parA$ $P_{Zn}$::$P_{Zn}$-parA-venus) were grown in MB medium supplemented with 0.3 mM $ZnSO_4$, transferred to an MB-agarose pad containing 0.3 mM $ZnSO_4$, and imaged at 30 min intervals by DIC and fluorescence microscopy. Bar: 3 µm. **c** Demographic analysis of ParA-Venus localization in swarmer (left), stalked (middle), and budding (right) cells. Cells of strain AJ46 were grown in MB medium containing 0.3 mM $ZnSO_4$ and analyzed by DIC and fluorescence microscopy. The fluorescence intensity profiles obtained for each cell type were then sorted according to cell length and stacked on top of each other

size, a bright focus remained associated with the old bud pole, but a fraction of the molecules returned to the mother cell, where they formed a variable number of polar and non-polar patches. Notably, there was no fluorescence detectable in the stalk at any stage of the cell cycle, suggesting that ParA may not contribute to the second segregation step (i.e., the transport of the *ori* region through the stalk).

To correlate the localization pattern of ParA with the dynamics of origin segregation, we visualized ParA-Venus and ParB-Cerulean in the same cells (Supplementary Fig. 2). At the beginning of the cell cycle, both proteins consistently colocalized at the old pole of the swarmer cell, suggesting a close interaction between them. Concomitant with the onset of origin replication and segregation, ParA-Venus then transitioned to a patchy distribution. Importantly, the relocation of ParA-Venus to the nascent bud occurred long before the initiation of the second segregation step, right at the onset of bud formation (also compare Fig. 2d). This process thus appears to be independent of the movement of the ParB·*parS* complex to the bud compartment, suggesting the existence of a thus-far unknown polar localization factor (analogous to TipN and PopZ in *C. crescentus*) that sequesters freely diffusible ParA molecules at the stalk-distal bud pole to control the dynamics of ParA-mediated *ori* segregation.

The above deletion and localization analyses suggest that ParA critically contributes to origin segregation. To further investigate its role in *H. neptunium*, we determined the phenotype of the conditional *parA* mutant (see Fig. 3a) after cultivation in non-inducing conditions. Due to leaky expression of the complementing *parA-venus* fusion (Fig. 4a), cells were still able to grow in the

absence of inducer. However, they developed severe morphological abnormalities, such as longer stalks and stretched, club-shaped bud compartments (Fig. 4b), leading to a considerable increase in their total cell length (Fig. 4c). To determine the distribution of DNA within these misshapen cells, we localized the nucleoids by DAPI staining. Cells producing ParA from the native or an inducible promoter consistently displayed DNA in both the mother and bud cell compartment once they had reached the late stages of the cell cycle. In the vast majority of ParA-deficient cells, by contrast, the nucleoid was restricted to the mother cell body, whereas the buds were free of DNA, indicating a DNA segregation defect (Fig. 4b, d). Flow cytometry revealed that, under these conditions, many cells contained three or more chromosome equivalents, whereas a maximum of two chromosomes were observed in the presence of a functional Par*ABS* system (Fig. 4e). Collectively, these results suggest that proper chromosome segregation is a prerequisite for efficient cell division and that DNA replication continues when cell division is blocked.

As protein depletion is a slow and often incomplete process, we aimed to establish a more rapid and efficient approach to block chromosome segregation. To this end, we introduced into ParA a dominant negative mutation (K18R) (Fig. 5a) that locks the protein in the ATP-bound, dimeric state[61,62] and thus actively blocks the movement of ParB·*parS* complexes across the nucleoid surface. We then localized ParB-Cerulean in a strain that ectopically expressed *parA*$_{K18R}$-venus under the control of a copper-inducible promoter (Fig. 5b). In the absence of inducer, most cells showed wild-type morphology and a normal DNA segregation pattern (Fig. 5c and Supplementary Fig. 3a). Induction of the mutant ParA fusion, by contrast, caused

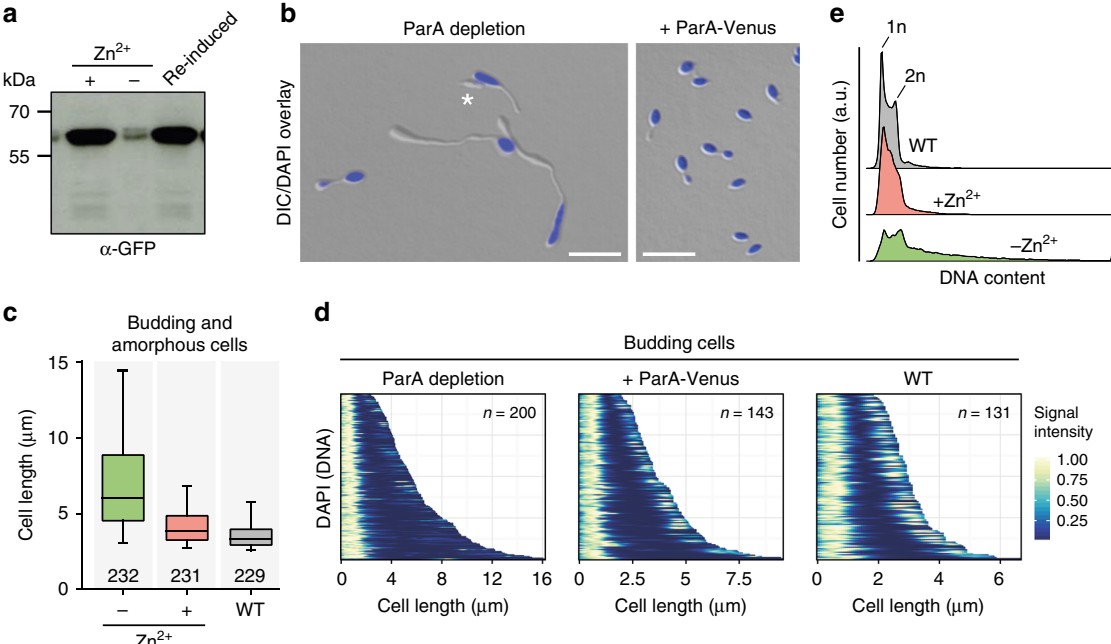

**Fig. 4** Effect of ParA depletion on chromosome partitioning. **a** Immunoblot showing the levels of ParA-Venus in strain AJ46 ($\Delta parA$ $P_{Zn}$::$P_{Zn}$-*parA-venus*) under inducing and non-inducing conditions. Cells were grown in MB medium containing inducer (0.3 mM ZnSO$_4$), washed, shifted to ZnSO$_4$-free medium, and cultivated for another 45 h to deplete ParA-Venus. Subsequently, cells were transferred to MB medium containing 0.5 mM ZnSO$_4$ and cultivated for 24 h to re-induce the synthesis of the fusion protein. Samples were subjected to immunoblot analysis with anti-GFP antibodies. A full scan of the immunoblot is shown in Supplementary Fig. 9A. **b** Microscopic analysis of cells depleted of ParA. Cells of strain AJ46 ($\Delta parA$ $P_{Zn}$::$P_{Zn}$-*parA-venus*) were grown in MB medium with or without 0.3 mM ZnSO$_4$ and incubated for 20 min with DAPI to stain the chromosomal DNA prior to imaging. Shown are overlays of DIC and fluorescence micrographs. The asterisk denotes an anucleate cell. Bars: 5 µm. **c** Box plots showing the length distributions of budding and amorphous cells from the cultures described in (**b**). The horizontal line indicates the median, the box the interquartile range, and the whiskers the 5th and 95th percentiles. The number of cells analyzed is given underneath the boxes. Wild-type cells grown in MB medium were analyzed as a control. Source data are provided as a Source Data file. **d** Demographic analysis of the distribution of chromosomal DNA in budding cells from the cultures described in (**b**). The fluorescence intensity profiles of random subpopulations of cells were stacked on top of each other according to cell length, with the mother cell body positioned to the left and the bud positioned to the right. **e** Flow cytometric analysis of the chromosome content of ParA-depleted cells. Strain AJ46 ($\Delta parA$ $P_{Zn}$::$P_{Zn}$-*parA-venus*) was grown in MB medium without or with 0.3 mM ZnSO$_4$. After staining of the chromosomal DNA with Vybrant DyeCycle Orange for 25 min, cells were subjected to flow cytometry. A wild-type culture grown in MB medium was analyzed as a control

morphological defects that initially were similar to those observed during ParA depletion. In addition, sister ParB·parS complexes were not or only barely separated and failed to move further to the bud compartment (Fig. 5c and Supplementary Fig. 3a–c). As observed above, DNA replication still continued under these conditions, leading to the accumulation of supernumerary chromosomes (Fig. 5d). After prolonged (18 h) induction of the mutant protein, cells developed highly abnormal shapes with multiple elongated stalks and amorphous bud compartments. Their swollen mother cell bodies displayed numerous ParB-Cerulean foci, whereas fluorescence was essentially undetectable in other cellular regions (Supplementary Fig. 3b). Collectively, these results demonstrate that ParA drives chromosome segregation in the mother cell, thereby positioning the moving *ori* region at the stalk base and enabling its subsequent translocation to the bud compartment.

**PopZ contributes to polar attachment of the origin regions**. The pole-organizing protein PopZ critically contributes to chromosome segregation in *C. crescentus*[35,36,43,44]. Searching for a similar factor in *H. neptunium*, we identified a thus-far uncharacterized protein (HNE_1677, now called PopZ$_{Hn}$) that showed 36% sequence identity with *C. crescentus* PopZ (PopZ$_{Cc}$). A particularly high degree of conservation was observed for the N- and C-terminal regions (Supplementary Fig. 4a), which were

shown to mediate the oligomerization and polar localization of PopZ$_{Cc}$ as well as its interaction with ParA and ParB[47,63,64]. In line with the identical genomic contexts of their genes (Supplementary Fig. 4b), we therefore concluded that the two proteins are true homologs.

To further investigate the role of PopZ$_{Hn}$ in *H. neptunium*, we constructed an inducible, fluorescently (Venus-) tagged variant of this protein (Supplementary Fig. 4c) and analyzed its localization dynamics by time-lapse microscopy (Fig. 6a) and demographic analysis of mixed cell populations (Fig. 6b). Swarmer cells consistently displayed a single focus at the old (flagellated) cell pole. After transition to the stalked phase, this pattern remained unchanged, until the stalk had reached its final length. Subsequently, the protein started to gradually relocate to the old pole of the nascent bud and stayed at this location up to the point of cell division. After the release of its daughter cell, the mother cell regenerated a new complex at the stalk tip, while initiating the formation of the next bud. The localization behavior of PopZ in *H. neptunium* thus clearly differs from the bipolar pattern observed in *C. crescentus*.

Colocalization studies revealed that PopZ$_{Hn}$ was closely associated with the ParB·parS complex in swarmer cells and, at a later stage of the cell cycle, in the bud compartment (Supplementary Fig. 5). This observation suggests that it may serve to sequester the segregated ParB·parS complex at the new

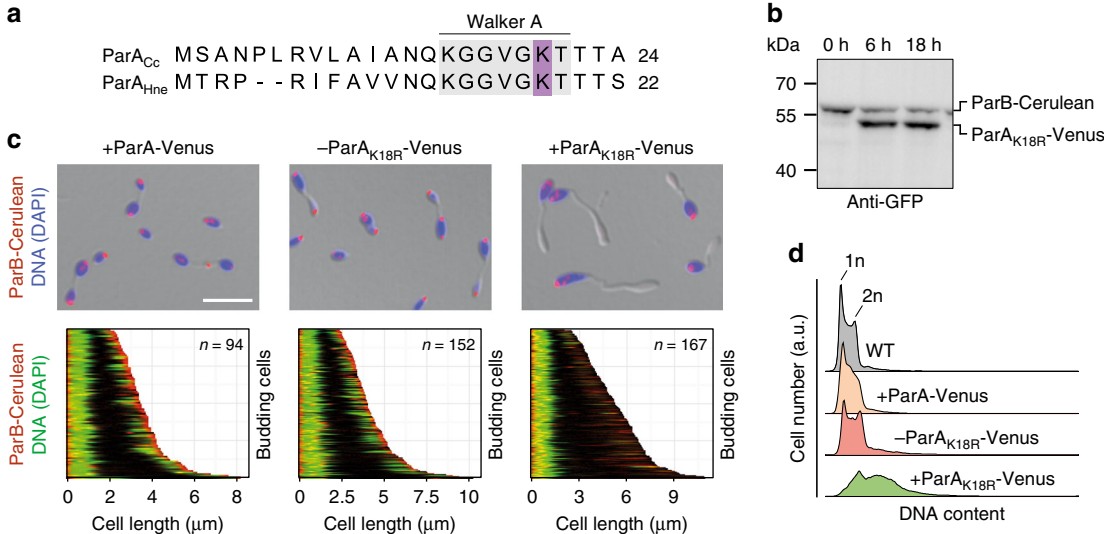

**Fig. 5** Effect of impaired ParA activity on *ori* segregation. **a** Alignment of the N-terminal amino acid sequences of ParA from *C. crescentus* and *H. neptunium*. The Walker A motif is highlighted in light gray. The conserved lysine residue mutated in this work is shown in purple. **b** Immunoblot showing the levels of ParA$_{K18R}$-Venus and ParB-Cerulean in strain AJ79 (*parB-cerulean* P$_{Cu}$::P$_{Cu}$-*parA$_{K18R}$-venus*) grown in the absence or presence of inducer. Cells were pre-grown in MB medium, induced with 0.5 mM CuSO$_4$, and cultivated for another 18.5 h. Samples were taken at the indicated time points and subjected to immunoblot analysis with anti-GFP antibodies. A full scan of the immunoblot is shown in Supplementary Fig. 9b. **c** Localization of ParB-Cerulean and chromosomal DNA in cells producing a dominant negative ParA variant. Strains AJ80 (*parB-cerulean* P$_{Cu}$::P$_{Cu}$-*parA-venus*) and AJ79 (*parB-cerulean* P$_{Cu}$::P$_{Cu}$-*parA$_{K18R}$-venus*) were pre-grown in MB medium and induced for 4.5 h with 0.3 mM CuSO$_4$. Subsequently, cells were treated for 25 min with DAPI to stain the nucleoids and subjected to microscopic analysis. A culture of strain AJ79 grown in the absence of inducer was analyzed as a control. Bar: 5 µm. The demographs at the bottom show the subcellular distribution of the ParB-Cerulean and DAPI signals in budding cells from the cultures described above, with the mother cell body positioned to the left and the bud positioned to the right. Black indicates the absence of specific signals. **d** Flow cytometric analysis of the chromosome content in cells producing a dominant negative ParA variant. Strains AJ80 (*parB-cerulean* P$_{Cu}$::P$_{Cu}$-*parA-venus*) and AJ79 (*parB-cerulean* P$_{Cu}$::P$_{Cu}$-*parA$_{K18R}$-venus*) were grown as described in (**d**) and treated for 25 min with Vybrant DyeCycle Orange to stain the chromosomal DNA prior to analysis (*n* = 30.000 cells)

bud pole and retain it there throughout the subsequent swarmer phase. To test this hypothesis, we generated an in-frame deletion in the corresponding gene. Microscopic analysis of the resulting mutant did not reveal any obvious morphological changes (Fig. 6c, d). However, in a significant fraction of the population (13.7%), the ParB·*parS* complex adjacent to the old mother cell pole was no longer located at the very tip of the cell but shifted towards the midcell region (Fig. 6e). Thus, PopZ$_{Hn}$ appears to be required for efficient anchoring of the *ori* regions but dispensable for chromosome segregation.

**The *H. neptunium* chromosome shows a longitudinal arrangement**. Our localization studies have so far focused on the origin-associated *parS* sites. To obtain a global view of the arrangement of DNA in *H. neptunium*, we aimed to determine the subcellular positions of ten different loci that were evenly distributed throughout the chromosome (Fig. 7a). For this purpose, we took advantage of the ParB-*parS* system of plasmid pMT1 from *Yersinia pestis*, which offers a straightforward means to fluorescently tag specific loci, based on a specific interaction between ParB$_{pMT1}$ with its cognate *parS*$_{pMT1}$ sites[13,31]. Initial analyses confirmed that there was no crosstalk between the endogenous and pMT1-encoded partitioning systems. We then constructed strains that carried *parS*$_{pMT1}$ sites at the loci of interest and produced a fluorescently (mCherry)-tagged derivative of ParB$_{pMT1}$ under the control of a zinc-inducible promoter. In addition, the native *parB* gene was replaced with a *parB-yfp* fusion to label the endogenous *parS* sites as a proxy for the chromosomal *ori* region. After imaging, we determined the subcellular positions of the *parS*$_{pMT1}$-tagged loci. In doing so, we

exclusively focused on swarmer and early stalked cells that displayed a single ParB-Venus focus, assuming that they are still in G1-phase and thus contain a single, fully replicated chromosome. Quantitative analysis of the localization data confirmed that the origin-associated ParB·*parS* complex was consistently localized near the old cell pole (Supplementary Fig. 6a, b). As the genetic distance from the replication origin increased, tagged loci gradually shifted towards the opposite end of the cell, with the *ter* region located in the immediate vicinity of the new pole (Fig. 7b). The *H. neptunium* chromosome thus appears to be organized into a ring-like or elongated structure, in which the subcellular location of loci roughly corresponds to their position on the circular chromosomal map.

**Both segregation steps initiate at the origin region**. After having clarified the arrangement of chromosomal DNA in G1-phase cells, we analyzed the dynamics of individual loci over the course of the replication cycle. To this end, we initially acquired images of mixed cultures and then determined the subcellular positions of tagged loci in cells at different stages of their developmental program. This analysis showed that origin-proximal segments of the chromosome (357° and 5°) were segregated shortly after replication initiation and closely followed the origin-associated ParB-*parS* complexes (Supplementary Fig. 6c). More distal loci, by contrast, stayed largely fixed in the midcell area, and their copies remained closely associated with each other until one of them was moved to the bud cell compartment. This observation is consistent with the previous finding that, in *H. neptunium*, chromosomal DNA is condensed into a clearly discernible

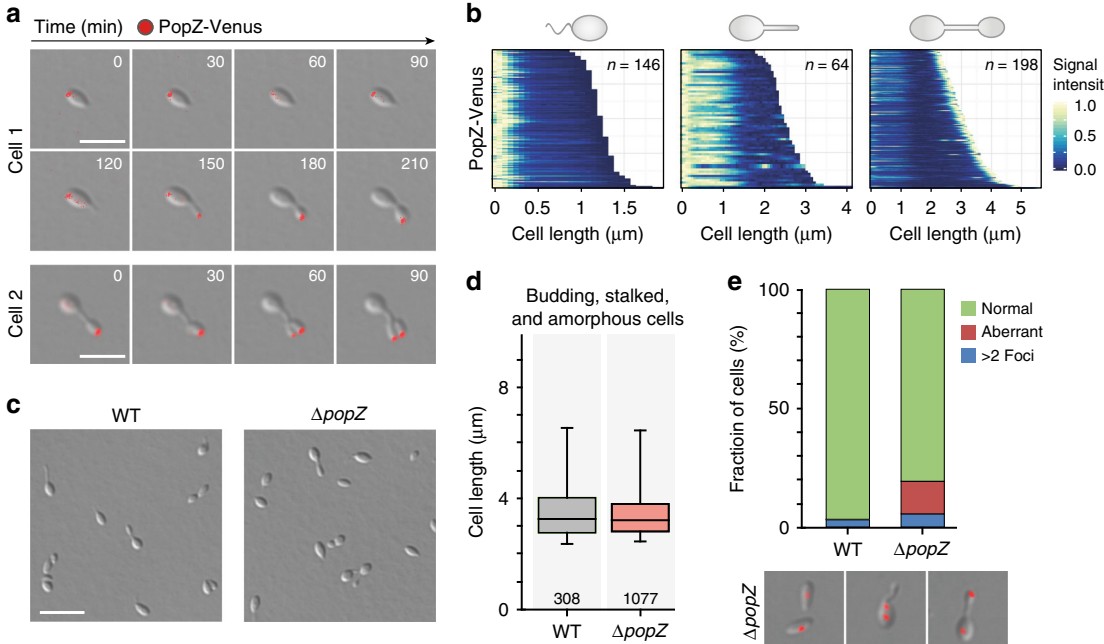

**Fig. 6 Role of PopZ in chromosome organization. a** Time-lapse series showing the localization of PopZ-Venus at different stages of the cell cycle. Cells of strain AJ34 ($P_{Zn}$::$P_{Zn}$-*popZ-venus*) were grown in MB medium, induced for 2.5 h with 0.5 mM $ZnSO_4$, transferred to an MB-agarose pad, and imaged at regular intervals. Shown are overlays of DIC and fluorescence micrographs. Bar 3 μm. **b** Demographic representation of PopZ-Venus localization in swarmer (left), stalked (middle), and budding (right) cells of strain AJ34 ($P_{Zn}$::$P_{Zn}$-*popZ-venus*) grown as described in (**a**). **c** Phenotype of a Δ*popZ* mutant. Cells of strain AJ38 (Δ*popZ*) and the *H. neptunium* wild type were grown in MB medium and visualized by DIC microscopy. Bar: 5 μm. **d** Distribution of cell lengths in the cultures described in (**c**). The data are represented as box plots. The horizontal line indicates the median, the box the interquartile range, and the whiskers the 5th and 95th percentiles. The number of cells analyzed is given underneath the boxes. Source data are provided as a Source Data file. **e** Comparison of ParB-YFP localization in wild-type and Δ*popZ* cells. Cells of strains AR48 (*parB-yfp*) and AJ89 (Δ*popZ parB-yfp*) were grown in MB medium and analyzed by DIC and fluorescence microscopy. The graphs show the proportion of cells with abnormal ParB-YFP localization patterns ($n = 534$ cells for AR48 and 612 cells for AJ89). Source data are provided as a Source Data file. The micrographs at the bottom give examples of the different categories quantified. Shown are overlays of DIC and fluorescence micrographs. Bar: 3 μm

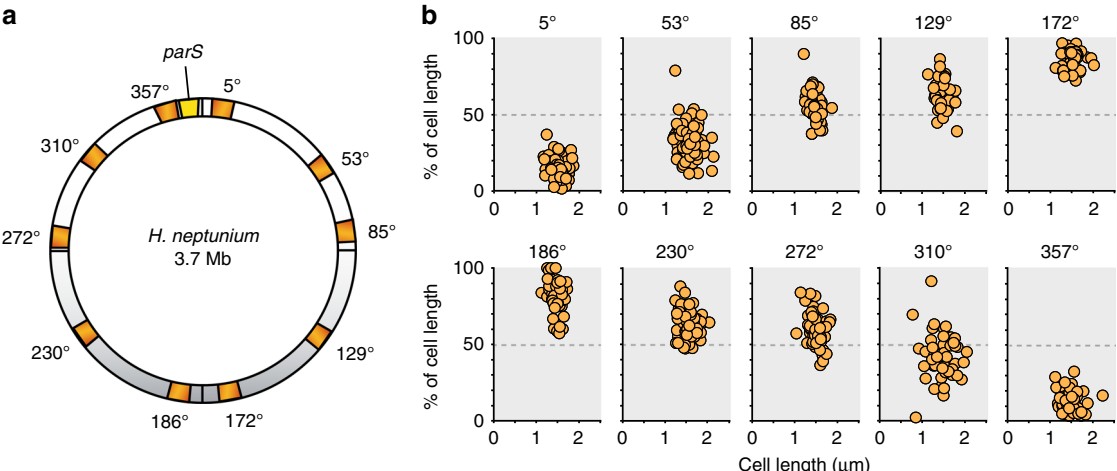

**Fig. 7 Spatial arrangement of the *H. neptunium* chromosome. a** Schematic showing the locations of chromosomal loci tagged with the *Y. pestis* ParB-*parS*$_{pMT1}$ system. **b** Subcellular localization of ten different chromosomal loci distributed evenly across the *H. neptunium* chromosome. Strains (*parB-yfp* $P_{Zn}$::$P_{Zn}$-*mCherry-parB*$_{pMT1}$) carrying *parS*$_{pMT1}$ at the indicated chromosomal locations (AJ64-69, SRE13-15, AJ49) were grown in MB medium and analyzed by DIC and fluorescence microscopy. The relative subcellular positions of the tagged loci (as reflected by the centers of mass of the respective foci) were determined in swarmer and early stalked G1-phase cells and plotted versus the total cell length, with 0% corresponding to the old (flagellated) pole and 100% to the new (future stalked) pole. $n = 75$ cells (5°), 118 cells (53°), 60 cells (85°), 48 cells (129°), 53 cells (172°), 50 cells (186°), 67 cells (230°), 64 cells (272°), 50 cells (310°), and 52 cells (357°)

nucleoid that occupies a distinct region at the cell center with less than 0.5 μm in diameter[51].

To gain deeper insight into the dynamics of chromosome segregation, we performed time-lapse analysis. However, mCherry-ParB$_{PMT1}$ only produced very faint signals, bleached rapidly and, surprisingly, was frequently stripped off the DNA once tagged loci entered the stalk. We therefore turned to the fluorescent repressor-operator system (FROS)[4,65] as an alternative in vivo labeling technique. Tandem arrays of the *lac* operator (*lacO*) were integrated at loci of interest in ParB-YFP-producing cells and detected with a fluorescently (mCherry-) labeled *lac* repressor. Time-lapse imaging showed that origin-proximal loci at 2° (~20 kb from *ori*) and 357° (~30 kb from *ori*) remained closely associated with *ori* throughout most of the cell cycle (Fig. 8a, b). However, ParB-YFP consistently moved ahead of the FROS label during both phases of origin segregation. A particular increase in the spacing of the two foci was observed during their passage through the stalk, suggesting that DNA is partially decondensed as it is transported from the mother cell to the bud compartment. These findings indicate that the *ori* region is the first segment of the chromosome to be moved during both the first and second segregation step.

To gain insight into the dynamics of origin-distal loci, we imaged strains carrying FROS labels at 53° and 172° (labeling the *ter* region), respectively. The two copies of the labeled segment were poorly separated as long as they resided in the mother cell and largely stayed in the subcellular region that the original locus occupied at the point of replication (Fig. 8c, d). Interestingly, while the FROS signal was normally unaffected by chromosome segregation, it was consistently lost from the chromosomal copy that moved to the bud when associated with the *ter* region. Thus, *ter* segregation appears to be mediated by a distinct mechanism that leads to the displacement of LacI-mCherry from its binding sites. In all cases, one of the two *ter* copies was retained at the stalk base (Fig. 8d and Supplementary Fig. 7), indicating that chromosome decatenation occurs within the mother cell and not, as usually observed[7], at the cell division site.

**The second segregation step initiates late in S-phase.** Our initial work suggested that the second step of *ori* segregation may initiate shortly before or after the end of S-phase (see Fig. 2a). To investigate this possibility in more detail, we aimed to clarify the timing of chromosome replication and its correlation with the segregation process. As a first approach to detect replication activity, we incubated growing cells with the thymidine analog 5-ethynyl-2′-deoxyuridine (EdU)[66,67] and visualized its incorporation into newly synthesized DNA by reaction with a fluorescent dye. EdU signals were barely detectable before the onset of stalk formation but highly prevalent in stalked cells (Fig. 9a). Their frequency then dropped again considerably in budding cells, indicating that S-phase starts shortly after the swarmer-to-stalked cell transition and likely ends well before the budding process is finished. EdU labeling studies in a strain producing ParB-YFP as a marker of the chromosomal origin regions further revealed that DNA synthesis initiated close to the chromosomal *parS* sites in early stalked cells and then gradually progressed through the mother cell body until it terminated at the stalk base in budding cells (Fig. 9b). A fraction of budding cells did not show any replication activity, whereas others displayed a signal that was again adjacent to the ParB-YFP focus in the mother cell body, which may reflect the reinitiation of chromosome replication right before cytokinesis.

To better resolve the cell cycle timing of chromosome replication, we monitored the dynamics of replisome assembly and progression in live cells. To this end, we generated a strain that carried a *dnaN-venus* fusion in place of the endogenous

*dnaN* gene (Supplementary Fig. 8), thus producing a fluorescently tagged derivative of the β-sliding clamp subunit of the DNA polymerase III holoenzyme[68]. Time-lapse analysis showed that, up to the swarmer-to-stalked-cell transition, DnaN-Venus was evenly distributed throughout the mother cell body (Fig. 9c). In early stalked cells, the protein condensed into a focus at the pole opposite the stalk. It then migrated slowly towards the stalked pole and finally dispersed some time after the onset of bud formation. Upon cell division, mother cells immediately initiated the next replication cycle, whereas the swarmer progeny again displayed an extended G1-phase. The directional pole-to-pole movement of DnaN-Venus (Fig. 9d and Supplementary Movie 1) suggests that the replisome tracks along the two adjacent arms of the chromosome, following the longitudinal arrangement of loci within the mother cell body (compare Fig. 7b). Notably, the moving focus frequently split into two distinct signals. The two replication forks thus appear to move independently of each other, although they may mostly remain in close proximity, so that it is difficult to resolve them individually by widefield microscopy. Collectively, these findings point to a dynamic localization of the replisome in *H. neptunium*, and they support the notion that DNA replication may be limited to the stalked and early budding stages of the cell cycle.

Having established a tool to precisely monitor the duration of S-phase, we aimed to shed light on the interplay between DNA replication and segregation. For this purpose, we followed the dynamics of the replisome and the chromosomal *ori* regions in the same cells, using a strain that produced both DnaN-Venus and ParB-Cerulean instead of the respective native proteins. Consistent with the results obtained by EdU labeling (see Fig. 9b), we observed that replisome assembly occurred in the immediate vicinity of the *parS* sites (Fig. 9e). Shortly afterwards, the *ori* region was duplicated and one of its copies started to migrate toward the stalked cell pole, suggesting that the two sister chromatids do not cohere for significant periods of time. Replication continued while the *ori* regions were fully partitioned to opposite poles of the mother cell and, after an extended waiting time, finally moved through the stalk into the bud compartment (Fig. 9f). At the onset of the second segregation step, DnaN-Venus was in most cases localized in the stalk-proximal half of the mother cell body, indicating that a large part of the chromosome had already been duplicated. The replisome finally disassembled ~40 min after the moving *ori* region had appeared in the bud (Fig. 9g). With a total replication time of ~118 min, as reflected by the interval between the formation and disintegration of the DnaN-Venus focus, the second segregation step thus initiates in the last third of S-phase. Collectively, these results show that there is a considerable delay between the synthesis and the final segregation of the two sister chromosomes in *H. neptunium*.

## Discussion

This study for the first time analyzes the arrangement and dynamics of chromosomal DNA in a stalked budding bacterium, using *H. neptunium* as a newly established model system. It shows that the unusual mode of proliferation observed for this species goes along with a novel pattern of chromosome segregation that largely uncouples the replication of DNA from its partitioning to the mother and bud cell compartments, reminiscent of eukaryotic mitosis.

Localization studies of ten different loci showed that the *H. neptunium* chromosome is organized into a ring-like structure, in which the *ori* and *ter* regions are located at opposite cell poles. Other loci, by contrast, are arranged sequentially along the long axis of the cell, with their subcellular locations roughly

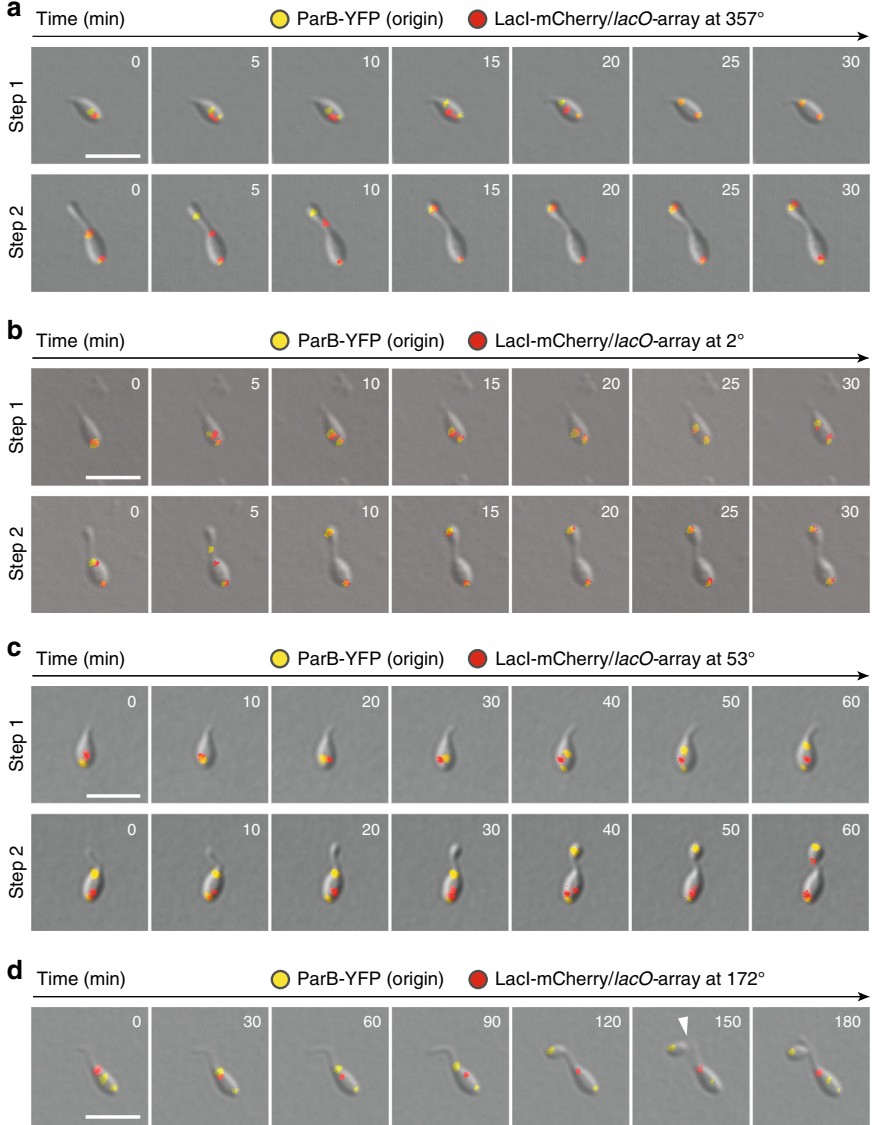

**Fig. 8** Dynamics of chromosome segregation. **a–d** Time-lapse series showing the segregation of a FROS-labeled chromosomal loci positioned at varying distance from the replication origin. Cells of strains **a** AJ87 (*HNE_3540::lacO*n *parB-yfp* P$_{Zn}$::P$_{Zn}$-*lacI-mCherry*), **b** JR50 (*HNE_HNE_0032::lacO*n *parB-yfp* P$_{Zn}$::P$_{Zn}$-*lacI-mCherry*), **c** KH24 (*HNE_0552::lacO*n *parB-yfp* P$_{Zn}$::P$_{Zn}$-*lacI-mCherry*), and **d** AJ86 (*HNE_1729::lacO*n *parB-yfp* P$_{Zn}$::P$_{Zn}$-*lacI-mCherry*) were induced for 1.5–3 h with 0.3 mM ZnSO$_4$, transferred to an MB-agarose pad, and imaged at regular intervals. Shown are overlays of DIC and fluorescence images. Bars: 3 µm

corresponding to their positions on the circular chromosomal map. A similar overall arrangement has been observed for several other bacterial species, including *C. crescentus*, *Myxococcus* xanthus, *Vibrio* cholerae and, to some degree, also *Pseudomonas aeruginosa*[4,10–12]. As a common feature, all of these organisms possess a ParABS system that drives active *ori* segregation. The longitudinal organization of their chromosomes may thus result from the polar positioning of the *ori* regions combined with the sequential recondensation of newly synthesized chromosomal segments, which are stacked upon each other in the order in which they emerge from the replication machinery.

Interestingly, unlike other species studied to date, *H. neptunium* segregates sister chromosomes in two distinct steps (Fig. 10). First, the *ori* regions are partitioned to opposite ends of the mother cell body, in a process resembling *ori* segregation in other model species. In a second step that has evolved specifically in *H. neptunium* cells, one of the two *ori* copies is then further transported through the stalk and finally positioned at the

stalk-distal pole of the bud compartment. Notably, this second step is delayed until close to the end of S-phase. The resulting lag between the first and second segregation step may serve two main purposes. On the one hand, it may allow the bud compartment to reach a size that is sufficient to accommodate the new nucleoid. On the other hand, it may ensure that the two sister chromosomes remain juxtaposed for some time to facilitate DNA repair. Recent work has shown that double-strand breaks induce a search of the damaged DNA segments for their intact counterparts on the sister chromosome to enable their repair by homologous recombination[69,70]. This process may become impossible once the sister chromatids have been partitioned to opposite sides of the stalk, because the small diameter of this structure likely prevents the efficient exchange of chromosomal regions between the mother and daughter cell compartments. It will be interesting to determine whether the small diameter of the stalk or its constriction during cell division could, to some extent, also hinder the diffusion of other macromolecules, such as proteins and

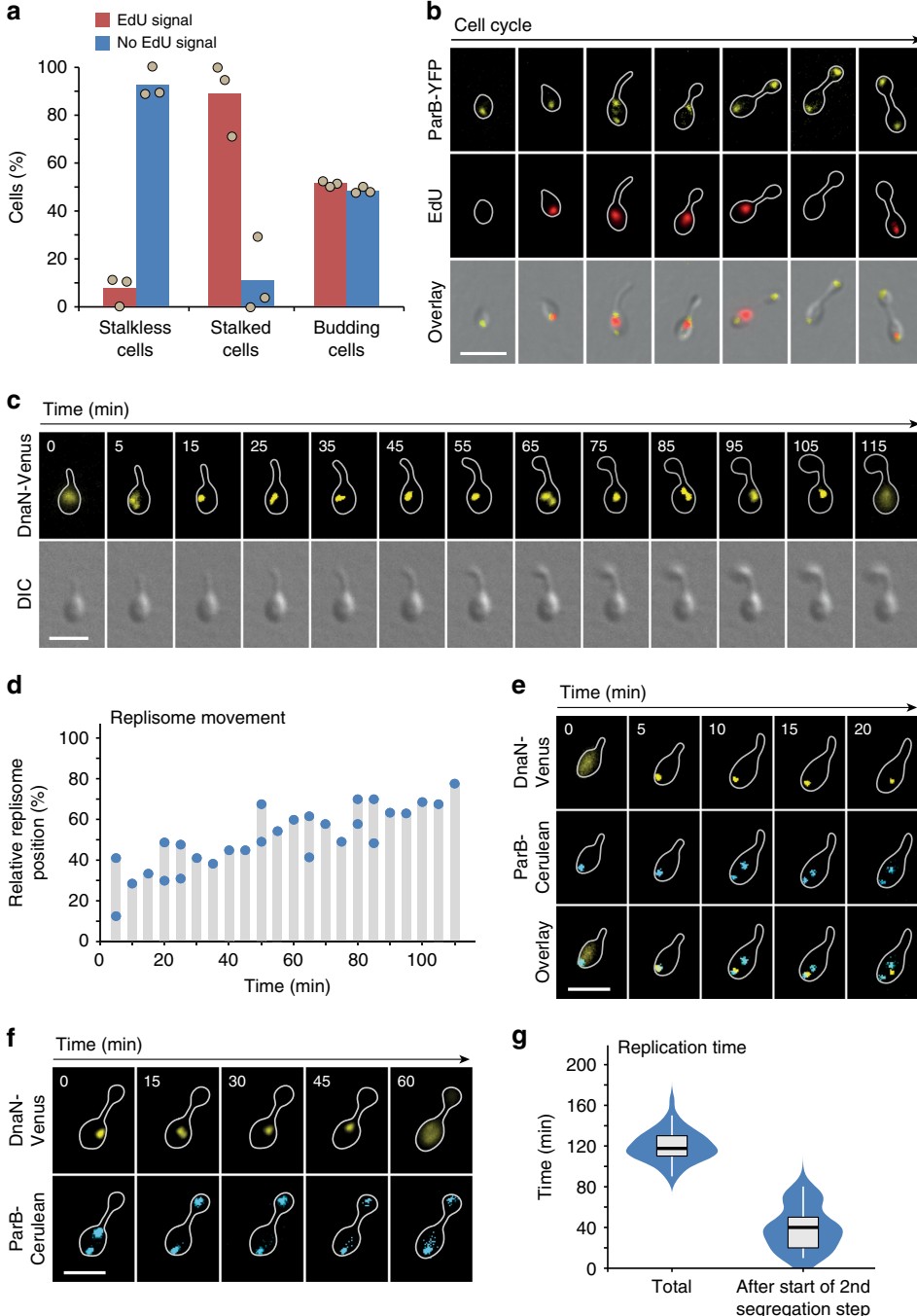

**Fig. 9** Dynamics of DNA replication in *H. neptunium*. **a** Quantification of S-phase cells at different stages of the developmental cycle. Exponentially growing wild-type cells were exposed to a pulse of the nucleotide analog EdU. After fixation and derivatization of the incorporated EdU with the red fluorescent dye Alexa 954, images were taken and the proportion of cells showing a fluorescence signal was determined for each of the indicated categories. Data represent the average of three experiments. **b** Localization of newly replicated DNA in *H. neptunium*. Strain KH22 (*parB-yfp*) was labeled with EdU as described in (**a**). Shown are representative images of cells at different developmental stages. Bar: 3 µm. **c** Time-lapse series showing the movement of the replisome along the two chromosomal arms. Cells of strain RP4 (*dnaN-venus*) were grown in MB medium, transferred to a 25% MB-agarose pad, and imaged at 5 min intervals. Only selected frames are presented. The full time-lapse series is shown in Supplementary Movie 1. Bar: 2 µm. **d** Quantitative analysis of replisome movement. The graph indicates the relative subcellular locations of the DnaN-Venus signals in the different frames of Supplementary Movie 1, with 0% indicating the old (previously flagellated) pole and 100% the future stalked pole of the mother cell. For frames in which the two replication forks were clearly separated, the positions of both DnaN-Venus foci are given. **e** Colocalization of the replisome with the ParB·parS complex at the beginning of S-phase. Strain JR47 (*dnaN-venus parB-cerulean*) was grown in MB medium, transferred to a 25% MB-agarose pad, and imaged at 5 min intervals. Bar: 2 µm. **f** Time-lapse series showing the dynamics of *ori* and replisome movement towards the end of S-phase. Cells of strain JR47 (*dnaN-venus parB-cerulean*) were grown in MB medium, transferred to a 25% MB-agarose pad, and imaged at 15 min intervals. Bar: 2 µm. **g** Quantification of the total replication time (n = 50 cells) and the interval between the start of the second segregation step and replisome disassembly (n = 30 cells). The data were obtained by analysis of the time-lapse series described in (**d**) and (**f**) and represented as box plots. The horizontal line indicates the median, the box the interquartile range, and the whiskers the 5th and 95th percentiles

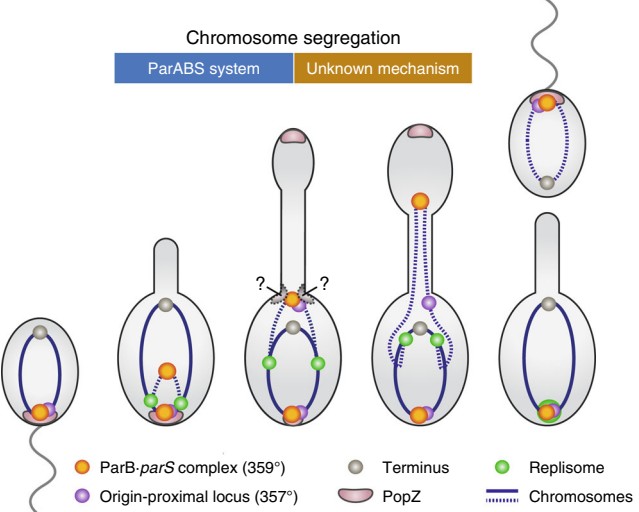

**Fig. 10** Model of the two-step chromosome segregation process in *H. neptunium*. In swarmer cells, the ParB·*parS* complex interacts with the scaffolding protein PopZ, thus anchoring the *ori* region at the flagellated pole. At the onset of S-phase, the replisome assembles at *ori* and the two replication forks start to move independently along the two arms of the chromosome. Soon after the initiation of replication, the two sister *ori* regions are segregated within the mother cell body, driven by the ParABS chromosome partitioning system. One of the two *ori* regions is immobilized at the stalk base, potentially through interaction of ParB with a thus-far unknown landmark protein. Once the replication forks have reached the last third of the chromosome, the second segregation step starts and the stalled *ori* region is translocated through the stalk to the bud compartment. After attachment of the ParB·*parS* complex to the flagellated bud pole and translocation of the remaining bulk of chromosomal DNA, the cell divides. The resulting stalked cell immediately starts the next replication cycle, whereas the swarmer cell first has to differentiate into a stalked cell to enter S-phase

RNAs. Our analyses show that the translocation of DNA into the nascent bud occurs over an extended period of time, with loci arriving sequentially according to their order on the chromosome. It is tempting to speculate that this process could establish a fixed temporal pattern of gene expression that helps to control bud development, provided that the gene products are retained in the bud compartment. A similar mechanism has previously been shown to establish a transient state of genetic asymmetry in sporulating *Bacillus subtilis* cells that is critical for proper spore formation[71–73]. Remarkably, a block in chromosome segregation leads to the generation of anucleate buds that fail to separate from the mother cell and continue to grow, eventually adopting a highly elongated, club-shaped morphology. This observation suggests the existence of a checkpoint that only licenses cell division once the second segregation step has finished, thus ensuring the formation of viable offspring. The precise mechanism driving the translocation of chromosomal DNA through the stalk is still unclear. Notably, we observed that DNA-associated proteins such as the chromosomal labels mCherry-ParB$_{pMT1}$ and, to some extent, LacI-mCherry are displaced from their binding sites once their target sites enter the stalk lumen. This phenomenon may be explained by changes in DNA topology caused by the decondensation and stretching of the chromosome during the translocation process. It could also hint at the involvement of a AAA$^+$ family DNA translocase such as FtsK or SpoIIIE, which form powerful motors that readily displace other proteins as they move along their DNA substrates[74,75]. *H. neptunium* contains a

predicted FtsK homolog (HNE_3541), and it will be interesting to determine whether this protein is involved in the second *ori* segregation step.

*H. neptunium* requires the ParABS system to properly segregate the *ori* regions within the mother cell body, with inactivation of ParA leading to severe morphological defects and cell death. Only a few other species, such as *C. crescentus*[29,76] and *M. xanthus*[10,77], have so far been shown to depend on active origin segregation. Many other organisms, by contrast, are able to survive in the absence of a functional ParABS system, although they often produce a small to moderate proportion of anucleate cells under this condition[11,37,78–83]. In some of them, the SMC complex emerged as a central player in the partitioning of sister chromosomes[84–86]. Notably, *H. neptunium* also contains an SMC homolog (HNE_1917), which appears to be required for viability (unpublished). This protein may contribute to sister chromosome segregation within the mother cell, but its precise role remains unclear. The essentiality of ParA and ParB in *H. neptunium* likely results from the need to position the *ori* region at the stalk base to set the basis for its subsequent translocation through the stalk. Although the *parS*-containing region is the first segment of the chromosome to be moved to the bud, ParA is unlikely to have a direct role in this process. The DNA partitioning activity of ParABS systems is known to require non-specific chromosomal DNA as a scaffold to facilitate the formation of the ParA gradient that directs the segregation reaction[35,36,40,47,87]. However, the stalk is devoid of DNA before the onset of the second segregation step, implying that the translocation of chromosomal DNA to the bud compartment is mediated by a novel, thus far unknown mechanism. It remains to be determined whether the underlying machinery requires ParB to initiate the translocation reaction or whether it can act on any stretch of DNA positioned within its reach at the stalk base. The events that occur once the moving *ori* region arrives in the bud are still unclear. Our results show that, after its entry, the ParB·*parS* complex traverses the bud compartment and attaches to the pole opposite the stalk. Since ParA relocates to the bud before the initiation of the second segregation step and later colocalizes with ParB at the stalk-distal bud pole, this process could again be mediated by the ParABS system. However, it may only occur once sufficient DNA has accumulated in the bud to support ParA gradient formation.

The complex pattern of *ori* movement in *H. neptunium* suggests the existence of a regulatory mechanism that controls the directionality and timing of this process. Previous work has shown that polar landmark proteins have a central role in the control of chromosome dynamics. On the one hand, they serve as anchors that tether the *ori* regions at the cell poles, thereby ensuring the proper arrangement of chromosomal DNA within the cell and its robust segregation during cell division[41,43–45,88–90]. On the other hand, they can also have a direct effect on the segregation reaction by interacting with ParA and controlling its activity state and localization pattern[36,45,90–92]. Two out of the previously identified landmark proteins are conserved in *H. neptunium*, including the polymer-forming proteins bactofilin and PopZ. We found that the chromosome segregation machinery is still positioned correctly in an *H. neptunium* bactofilin mutant (unpublished). Inactivation of PopZ, by contrast, compromised the polar attachment of the *ori* region during G1-phase, but it did not cause any other obvious defects in *ori* segregation, growth or cell division. Unlike in *C. crescentus*[36,47,91] and *Agrobacterium tumefaciens*[42,93,94], PopZ may thus not play a major role in the regulation of chromosome partitioning in *H. neptunium*. This finding suggests the existence of thus-far unknown factors that mark the four poles of the *H. neptunium* cell to enable the proper targeting of cellular components during the different stages of its complex developmental cycle.

Apart from the events involved in chromosome segregation, we also clarified the dynamics of DNA replication in *H. neptunium*. Our results show that the replisome assembles during the swarmer-to-stalked cell transition and then gradually moves from its starting point at the previously flagellated pole towards the stalked pole of the mother cell, where it finally disassembles. A similar behavior was observed in *Escherichia* coli, *M. xanthus*, and *C. crescentus*[10,95,96], whereas the *P. aeruginosa* replisome remains largely stationary in the midcell region[11]. Notably, the replisome frequently split into two distinct complexes, supporting the notion that the two replication forks move independently of each other[10,60,96]. *H. neptunium* thus uses a conserved set of mechanisms to duplicate and segregate chromosomal DNA within the mother cell body. However, additional systems have been put in place to adapt the segregation process to the specific needs of this species.

The study of stalked budding bacteria provides insight into the fascinating diversity of the mechanisms that have evolved to spatiotemporally organize bacterial cells. It will be interesting to identify the polar landmarks that may tether the *ori* region to the stalk base and control the directionality of the chromosome segregation process. Another important question concerns the cue that triggers the second segregation step and thus coordinates chromosome segregation with the developmental cycle of *H. neptunium*. Finally, studies of the mechanism that mediates the translocation of chromosomal DNA through the stalk will add to our understanding of the molecular principles controlling the dynamics of chromosomal DNA in bacteria.

## Methods

**Growth conditions**. *H. neptunium* LE670 (ATCC 15444) and its derivatives were grown in Difco Marine Broth 2216 (MB) medium (BD Biosciences, Germany) at 28 °C under aerobic conditions (shaking at 210 rpm) in baffled flasks. When appropriate, media were supplemented with antibiotics at the following concentrations (μg ml$^{-1}$ in liquid/solid medium): rifampicin (1/2), triclosan (0.25/0.25), kanamycin (100/200). The expression of genes controlled by the $P_{Cu}$ or $P_{Zn}$ promoters[52] was induced by addition of CuSO$_4$ or ZnSO$_4$, as described in the text. To promote stalk elongation, *H. neptunium* was incubated in phosphate-limited minimal medium[97]. *E. coli* was cultivated in LB medium (shaking at 210 rpm) at 37 °C. For plasmid-bearing strains, antibiotics were added at the following concentrations (μg ml$^{-1}$ in liquid/solid medium): kanamycin (30/50), rifampicin (25/50), ampicillin (50/200), triclosan (2/2). To grow *E. coli* WM3064, media were supplemented with 2,6-diaminopimelic acid (DAP) at a final concentration of 300 μM. To assess the growth of *H. neptunium*, cells were grown to exponential phase, diluted with fresh medium to an optical density at 580 nm (OD$_{580}$) of 0.025, and transferred into 24-well polystyrene microtiter plates (Becton Dickinson Labware, USA). Growth was then followed at 32 °C under double-orbital shaking in an EPOCH 2 microplate reader (BioTek, USA) by measuring the OD$_{580}$ at 30 min intervals.

**Plasmid and strain construction**. The bacterial strains, plasmids, and oligonucleotides used in this study are listed in Supplementary Tables 1–4. *E. coli* TOP10 (Invitrogen) was used as host for cloning purposes. All plasmids were verified by DNA sequencing. *H. neptunium* was transformed by conjugation[52]. Non-replicating plasmids were integrated into the *H. neptunium* chromosome by single-homologous recombination at the $P_{Cu}$ or $P_{Zn}$ locus[52]. Gene replacement was achieved by double-homologous recombination using the counter-selectable *sacB* marker[51]. Proper chromosomal integration or gene replacement was verified by colony PCR.

**Flow cytometry**. Cultures were grown to mid-exponential phase, diluted to an OD$_{600}$ of 0.1–0.2, and incubated under vigorous shaking for 25 min with the DNA-specific fluorescent dye Vybrant DyeCycle Orange (Invitrogen, Germany) at a final concentration of 10 μM. Subsequently, cells were analyzed by flow cytometry in a customized Fortessa Flow Cytometer (BD Biosciences), using an excitation wavelength of 514 nm and a Blue 530/30 band-pass emission filter. Data were acquired with FACSdiva 8.0 (BD Biosciences) and processed with FlowJo v10 (FlowJo LLC).

**Marker frequency analysis**. Swarmer cells isolated from an *H. neptunium* wild-type culture were cultivated for 40 min to obtain a synchronous population that has just entered S-phase. In parallel, a mixed culture was grown to stationary phase (OD$_{600}$ = 1.4) and thus enriched for cells arrested in a non-replicating state. Two samples were taken from each culture and used to extract chromosomal DNA with

the NucleoSpin Microbial kit (Macherey-Nagel, Germany). Libraries were prepared from each of the four DNA samples using the Nextera XT Library Preparation Kit (Illumina, USA). Sequencing was performed using a MiSeq v3 Reagent Kit with 150 cycles on a MiSeq System (Illumina, USA). At least $1.8 \times 10^6$ paired-end reads of $2 \times 75$ nucleotides length were obtained per library. To analyze the frequency of chromosomal markers, we mapped the reads to the *H. neptunium* reference genome[50] and calculated for both datasets the coverage of the total genome and the average number of reads for 1000 equally spaced 1 kb windows. Windows that did not properly reflect the average coverage of the individual bases or only partially mapped to the reference genome were removed. The values obtained for the synchronized culture were then corrected for sequence biases by using the data from the stationary-phase cells[98]. The results were further corrected by a local correction factor[98], derived from the two samples of the synchronized culture, to reduce intrinsic sequence biases.

**Microscopy and image analysis**. Exponentially growing cultures were used for all microscopic analyses. The synthesis of fluorescent protein fusions was induced as indicated. To visualize nucleoids, cells were incubated with 4 μg ml$^{-1}$ DAPI (4′,6-diamidino-2-phenylindole) for 20 min at 28 °C under vigorous shaking (400 rpm). To acquire single snapshots, cultures were spotted onto 1% agarose pads prior to imaging at room temperature. For time-lapse analysis, cells were immobilized on pads made of 1% agarose in MB medium. The cover slides were then sealed with VLAP (1:1:1 mixture of vaseline, lanolin, and paraffin) to prevent dehydration, and microscopy was performed in an Incubator XL-4 climate chamber (PeCon, Germany) adjusted to a constant temperature of 28 °C. Images were taken with a Zeiss Axio Observer.Z1 microscope equipped with an alpha Plan-Apochromat 100×/1.46 Oil DIC M27 and a Plan-Apochromat 100×/1.40 Oil Ph3 M27 objective (Zeiss, Germany). An X-Cite 120PC metal halide light source (EXFO, Canada) and ET-DAPI, ET-CFP, ET-YFP, or ET-TexasRed filter cubes (Chroma, USA) were used for fluorescence detection. Pictures were taken with a pco.edge sCMOS camera (PCO, Germany), recorded with VisiView 2.1.4 (Visitron, Germany), and processed with MetaMorph 7.7 (Universal Imaging, USA) and Adobe Illustrator CS5 (Adobe Systems, USA).

**EdU-click labeling of newly synthesized DNA**. Newly synthesized DNA was labeled with 5-ethynyl-2′-deoxyuridine (EdU)[99] using the Click-iT EdU Alexa Fluor Imaging Kit (Life Technologies, Germany). An exponentially growing culture was supplemented with 0.06–0.24 mM EdU. After 5–15 min, the reaction was stopped by fixation with ethanol or methanol (86% final concentration). The cells were harvested by centrifugation ($5000 \times g$, 4 °C, 5 min) and washed two times with 1× PBS. The final pellet was then resuspended in 200 μl Click-it reaction cocktail per 1 ml of cell culture and incubated for 30 min at room temperature. Afterwards, the cells were collected, washed, resuspended in 1× PBS, and analyzed by DIC and fluorescence microscopy.

**Immunoblot analysis**. Immunoblot analysis was performed according to standard procedures[52] using anti-GFP (Sigma-Aldrich, Germany; Product number G 1544) and anti-mCherry (Bio Vision, USA; Catalog number 5993-100) antibodies at dilutions of 1:10,000. Immunocomplexes were visualized with the Western Lightning Chemiluminescence Reagent Plus kit (PerkinElmer, USA) according to the manufacturer's instructions. Chemiluminescence was detected with Amersham Hyperfilm ECL films (GE Healthcare, Germany) or with a ChemiDoc MP imaging system (Bio-Rad, Germany). PageRuler Prestained 10–180 kD Protein Ladder (Thermo Fisher Scientific, USA) was used as a molecular weight standard.

**Protein purification**. In order to purify ParB (HNE_3560)-His$_6$, *E. coli* Rosetta (DE3)pLysS (Invitrogen, Germany) was transformed with plasmid pAJ40 and grown in LB supplemented with ampicillin (50 μg ml$^{-1}$) and chloramphenicol (20 μg ml$^{-1}$) to an OD$_{600}$ of 1. Protein overproduction was induced by addition of 0.5 mM IPTG. After 3 h of incubation at 37 °C, the cells were harvested, washed twice with buffer B2 (50 mM NaH$_2$PO$_4$, 300 mM NaCl, 10 mM imidazole; adjusted to pH 8.0 with NaOH). ParB-His$_6$ was purified by nickel affinity chromatography using an ÄKTApurifier 10 system (GE Healthcare, Germany). To this end, cells were resuspended in 10 ml Buffer B3 (50 mM HEPES/NaOH, 150 mM NaCl, 20 mM imidazole; pH 8.0) per 1 g of cell pellet, supplemented with 25 μg ml$^{-1}$ DNaseI, 100 μg ml$^{-1}$ PMSF (phenylmethylsulfonyl fluoride) and 20 μg ml$^{-1}$ lysozyme, and lysed by two passages through a French press (16,000 psi). After the removal of cell debris by centrifugation for 20 min at $4000 \times g$, the lysate was subjected to ultra-centrifugation at $160,000 \times g$ for 1 h (4 °C). The supernatant was then loaded on a 5 ml HisTrap HP column (GE Healthcare, Germany) equilibrated with buffer B3. After a wash with the same buffer, the protein was eluted using a linear (20–400 mM) imidazole gradient obtained by mixing buffer B3 with buffer B4 (50 mM HEPES/NaOH, 150 mM NaCl, 400 mM imidazole; pH 8.0). Fractions containing the protein of interest were pooled and diluted 1:3 with buffer A (50 mM HEPES/NaOH, pH 8.0). Subsequently, the solution was applied to a 1 ml HiTrap Q FF anion exchange column (GE Healthcare, Germany) equilibrated with buffer A. After a wash with the same buffer, protein was eluted with linear NaCl gradient (0–1 M NaCl) obtained by mixing buffer A with buffer B (50 mM HEPES/NaOH, 1 M NaCl; pH 8.0). Fractions containing pure ParB-His$_6$ were pooled and dialyzed

against buffer B6 (50 mM HEPES/NaOH, pH 7.2, 50 mM NaCl, 5 mM MgCl$_2$, 0.1 mM EDTA, 10% glycerol). The purified protein was aliquoted, snap-frozen in liquid nitrogen, and stored at −80 °C. Protein concentrations were determined spectrophotometrically with a modified Bradford assay using Roti®-Nanoquant (Carl Roth, Germany) according to the manufacturer's instructions. Proteins separated by SDS-PAGE were detected with Instant*Blue* (Merck).

**Electrophoretic mobility shift assay**. To test the ability of ParB-His$_6$ to bind the *parS* motif, 0.01 μM of a Cy3-labeled double-stranded DNA oligonucleotide containing either the wild-type (5′-TGTTTCACGTGAAACA-3′) or a mutated (5′-TGCCTCACGTGAAACA-3′) *parS* sequence were incubated with 0.05 μg μl$^{-1}$ poly(dI-dC) (Sigma Aldrich, Germany) and varying concentrations of ParB (0–0.6 μM) in buffer B6 (50 mM HEPES/NaOH, pH 7.2, 50 mM NaCl, 5 mM MgCl$_2$, 0.1 mM EDTA, 10% glycerol) for 30 min at 28 °C. 20 μl samples were mixed with 6× DNA loading dye (Fermentas, Germany) and loaded on a 6% non-denaturing polyacrylamide gel, which was run for 50 min at a constant voltage of 120 V and 4 °C in 1× TBE buffer. Signals were detected with a Typhoon 8600 imager (GE Healthcare).

**Bioinformatic analysis**. To generate demographs, fluorescence intensity profiles were measured with ImageJ 1.47v (http://imagej.nih.gov/ij). The data were then processed in R version 3.1.1 (The R Foundation for Statistical Computing; http://www.r-project.org) using the Cell Profiles script (http://github.com/ta-cameron/Cell-Profiles)[100]. Box plots were generated using QTI plot (http://www.qtiplot.com).

**Enrichment of *H. neptunium* swarmer cells**. *H. neptunium* cultures were enriched for swarmer cells by size-selective filtration[49,101]. To this end, *H. neptunium* was grown to an OD$_{600}$ of 0.6 in 300 ml MB medium. The cells were harvested by centrifugation (15 min, 3000 × *g*, 4 °C), resuspended in 100 ml 1× PBS, and vacuum-filtered through a nitrocellulose membrane with a pore size of 1.2 μm (Millipore, Germany). The flow-through was collected and then filtered again, using membranes with a pore size of 0.8 μm (Millipore, Germany). Swarmer and early stalked cells were collected from the flow-through by low-speed centrifugation (15 min, 3000 × *g*, 4 °C), resuspended in pre-warmed MB medium, and then cultivated at 28 °C prior to analysis.

**Reporting summary**. Further information on research design is available in the Nature Research Reporting Summary linked to this article.

## Data availability

Source data underlying Figs. 4c, 6d, e and Supplementary Fig. 3 are provided as Source Data file. The authors declare that the main data supporting the findings of this study are available within the article and its Supplementary Information files. All other data are available from the corresponding author upon reasonable request.

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

## Acknowledgements

The authors thank Julia Rosum, Stephanie Steede, and Bernadette Boomers for excellent technical assistance, Silvia Gonzales Sierra at the SYNMIKRO Flow Cytometry Facility for help with the flow cytometry analysis, Sven Reislöhner for the generation of strains, and Wolfgang Strobel for support in the initial phases of this study. Moreover, the authors are grateful to Manuel Osorio Valeriano, Laura Corrales Guerrero, and Maria Billini for critical reading of the manuscript. This work was supported by funding from the Max Planck Society (Max Planck Fellowship to M.T.), the LOEWE program of the State of Hesse (to P.S.), and the German Research Foundation (Project 269423233—TRR 174; to A.B. and M.T.). M.C.F.v.T. acknowledges support from the European Molecular Biology Organization (EMBO Long-Term Fellowship ALTF 1396–2015). R.L.P. is a fellow of the International Max Planck Research School for Environmental, Cellular and Molecular Microbiology (IMPRS-Mic).

## Author contributions

A.J., A.R., R.L.P., M.C.F.v.T., and K.H. performed the experiments and analyzed the data. P.S. performed the marker frequency analysis. J.S. and A.B. provided sequencing technology. A.J. and M.T. conceived the study. A.J., M.C.F.v.T., and M.T. wrote the paper, with input from all other authors.

## Additional information

**Competing interests:** The authors declare no competing interests.

**Peer Review Information:** *Nature Communications* thanks Diego Cattoni, Maria Schumacher, and other anonymous reviewer(s) for their contribution to the peer review of this work. Peer reviewer reports are available.

