## [Peer Review File · Nature Communications]

Reviewers' comments:

Reviewer #1 (Remarks to the Author):

This study by Jung at co-workers provides a detailed analysis of the arrangement and dynamics of chromosomal DNA in *Hyphomonas neptunium*, a stalked budding bacterium.

Interestingly, unlike the traditional models, *H. neptunium* segregates its sister chromosomes in two distinct steps. First, the origin regions (ParB/parS) are partitioned to opposite ends of the mother cell body, in a process likely to resemble origin segregation in well-studied model bacteria (mediated by ParA). In a second step that is thus far unique to *H. neptunium* cells, one of the chromosomal origins is transported through the stalk (followed by the rest of the chromosome) and positioned at the stalk-distal pole of the bud compartment. Of particular interest, this study nicely demonstrates that both steps initiate at or near the origin.

The manuscript is extremely thorough and covers a lot of ground, documenting the localization dynamics of ParB, ParA, PopZ, and chromosomal loci. The work also investigates the dynamics of DNA replication. Although the results are principally descriptive I think they provide new information that is unique to this unusual bacterium (or possibly bacteria in general that employ this mode of growth). Furthermore, the work raises as many new and interesting questions as it answers and lays the groundwork for more mechanistic studies in the future.

The writing is very clear and easy to follow - appropriate for a broad audience. Further, the quality of the data, especially the cytological analysis, is quite high and made it a pleasure to read (despite the presentation of so much data).

Below are some more minor concerns that I encourage the authors to address:

1) I would include topoisomerases as key players in chromosome segregation (with their ability to resolve rather than tangle DNA likely driven by the juxtaposition of chromosome "arms" by SMC complexes). Does *H. neptunium* have an SMC complex? Might be worth mentioning/discussing.

2) I found the analogy to eukaryotic mitosis ("reminiscent of eukaryotic mitosis" was mentioned in the Abstract, Intro, Results, and Discussion) not particularly compelling. While it is true that the 2nd

phase of segregation occurs after S-phase is mostly complete, the chromosomes are not cohesed at this stage as is the case in eukaryotes. Accordingly, the only similarity is the timing of segregation relative to replication. To my mind this process much more closely resembles spore formation in *B. subtilis* in which one of the replicated chromosome (or at least most of it) is transferred into the spore compartment after replication is complete during a late stage in cytokinesis.

3) The observation that one of the fluorescent ParB proteins (mCherry-ParBpMT1) used for FROS was stripped off the chromosome is reminiscent of the stripping function of FtsK observed in *B. subtilis* sporulation. I was surprised that the authors did not discuss a possible role for this DNA translocase in transporting the DNA through the stalk in the Discussion (I assume and FtsK is present in this bacterium). Especially because they show that a FROS signal associated with the *ter* region is consistently lost when it enters the stalk - suggesting that FtsK/XerCD dimer resolution occurs at the stalk base. Thus, FtsK would be well positioned to promote (or even drive) DNA transfer.

4) The observation that origin DNA enters the bud prior to the rest of the chromosome prompted the authors to speculate that this manner of chromosome segregation could establish a fixed temporal pattern of gene ex-pression that helps to control bud development. This would be a good place to mention the transient genetic asymmetry observed in *B. subtilis* during spore formation. The original description of this phenomenon comes from Patrick Stragier (Frandsen et al 1999) but the demonstration of its biological relevance comes from Patrick Piggot and A. Hofmeister (Zupancic 2001, Khvorova et al 2000). I like the idea that *H. neptunium* might similarly take advantage of transient genetic asymmetry - or at least place specific genes closer to the origin (than *C. crescentus* does) to ensure sufficient quantities are produced as soon as possible.

Reviewer #2 (Remarks to the Author):

General comment

In their manuscript Jung et al. combine bioinformatics analysis, biochemical approaches and microscopy to unveil the molecular actors and mechanisms regulating chromosome segregation in the *Hyphomonas neptunium* bacterium. Interestingly they report that in *H. neptunium* the segregation of the replicated chromosome occurs in a two-step manner. This is completely novel in bacteria and opens the door to new perspectives in the study of bacterial chromosomal dynamics. In some portions of the manuscript the authors emphasize that there is an uncoupling between DNA synthesis and chromosome segregation and I believe that this can not be unambiguously concluded

from their results. Although the hypothesis remains reasonable from the data, I am more inclined to interpret the results as a partial uncoupling. Overall, this study is very well executed, thoroughly controlled, and the conclusions drawn are sound.

I point below some questions/comments/suggestions for the authors hoping it will help to clarify some aspects of the manuscript.

Data analysis:

In all figures where the authors perform a demographic analysis of the localization of protein complexes the color coded scale is confusing. What does intensity means in those scales? Fluorescence intensity? Why did the authors choose not to find the center of mass for each foci (for ParB and PopZ)? This will give richer spatial information.

In figure 7 it is unclear if the same demographic strategy was employed to localize the position of chromosomal loci.

In figure 8 more quantitative information could be obtained, and perhaps easier to understand for a wider audience, if a plot of distance vs. time for each studied pair of loci was included. This will also allow to compare the velocity of displacement of each chromosomal region, if the distance between pairs of loci remains constant through cell cycle and perhaps to gain additional mechanistic insight.

In figure 2C panel 2 it appears as the cell displays two foci at the stalk. How do the authors explain this? Was this behavior present in a large majority of cells?

The authors report two parS sites for *H. neptunium*. Most bacteria and plasmids carrying a ParABS system contain several repeats of parS. Moreover, a recent study confirmed the existence of additional parS sites in *C. crescentus* (10.1093/nar/gkx1192). Did the authors explore the possibility of additional parS sites in *H. neptunium*?

In figure 5 panel D it was unclear what the black color indicates. I am not sure if colocalization is the correct way to call this analysis. The authors are not really interested in colocalization as DAPI signal is faint and distributed in the whole cell.

Manuscript organization:

The introduction may be shortened and more focus on the central aim of the paper, particularly the long paragraph devoted to *C. crescentus*. If comparison between the chromosome segregation mechanisms of *C. crescentus* and *H. neptunium* is pertinent I suggest to develop that in the discussion section.

Although I found all figures relevant I wonder if there could be a way to fusion some of them in order to make the article more compact and easy to follow for a wider audience.

Discussion:

The authors indicate that ParA, as it has been reported previously, requires the DNA scaffold to 'travel' and create a gradient. The authors address this in the discussion, however I found puzzling that in figure S2 on the fourth, fifth and sixth lines of panels (from the right, cell cycle progression) there is ParA signal on the bud while the two origins remain in the mother cell. How do the authors explain this?

The results of measurements of translocation speed in the stalk strongly argue for an active mechanism. In the discussion the authors suggest the existence of an unknown mechanism. Does *H. neptunium* possess any FtsK-like translocase gene that could be a candidate for such an active mechanism?

Does *H. neptunium* have SMC-like candidates?

Minor points:

The legend of Figure 1 has inverted the order of the panels C and D.

In all microscopy figures. Could the authors add an inset in each of them showing a zoomed individual and representative bacteria to be able to appreciate the foci size and shape (maybe with a dark background as they do in some other figures)? In all cases the bacteria images are very small and it is hard to see additional details.

Page 9, line 195. The figure called there should be 4E.

Could the authors include the full name of the microscopy technique the first time it appears mentioned in the text in legend of Figure 2 'Differential interference contrast (DIC) microscopy'?

The movie S1 was not present among the files received for the reviewing process.

Diego Cattoni

Reviewer #3 (Remarks to the Author):

The manuscript from the Thanbichler's lab interrogates a fundamental biological process that is surprising poorly understood in bacteria, which is the mechanism of chromosome segregation. Previous work by the Thanbichler lab revealed important insights into DNA segregation in the

related alpha proteobacteria *Caulobacter crescentus*. In this manuscript, the authors did a large amount of work focusing on the alphaproteobacteria *H. neptunium*, for which tools have only recently been developed, enabling the cellular dissection of its biological processes. This cell biological study reveals that the *parABS* partition system of *H. neptunium* is essential for DNA segregation and that it employs a two step process for segregation involving bud formation. Given how little we know about bacterial segregation, I believe this work represents an important contribution to the field. Therefore I am in favor of publication but have just a few questions to be addressed.

1. While the *H. neptunium* appears to have a PopZ, the analyzes suggests it is functionally different from the *C. crescentus* PopZ. But it was unclear to me from their studies whether it interacts with ParB as does the *Caulobacter* PopZ? do they see the protein colocalize?

2 . The authors say that the studies in Fig.7 show that the *H. neptunium* chromosome is circular. But it doesn't seem like these studies have the resolution to make that claim. Clearly the chromosome is not tightly condensed with Ori and Ter in the same volume for example, but the chromosome could still be compacted with a elongated, packed structure, based on the results.

3. Also, the image at 186 degrees (Fig. 7) seems to be missing the *parS* signal. Is that because the *parS* region is overlapping with the region of the chromosome?

4. On page 6 the authors say they generated a strain producing a fully functional ParB-YFP fusion in place of the native ParB. But what is the data showing that? Maybe I missed that.

5. In figure 1 the legend for C should be D and vice versa; the legends are reversed.

We thank all three reviewers for their positive view of our work, their constructive criticism and their suggestions on how to further improve the paper!

Reviewer #1 (Remarks to the Author):

This study by Jung at co-workers provides a detailed analysis of the arrangement and dynamics of chromosomal DNA in *Hyphomonas neptunium*, a stalked budding bacterium.

Interestingly, unlike the traditional models, *H. neptunium* segregates its sister chromosomes in two distinct steps. First, the origin regions (ParB/parS) are partitioned to opposite ends of the mother cell body, in a process likely to resemble origin segregation in well-studied model bacteria (mediated by ParA). In a second step that is thus far unique to *H. neptunium* cells, one of the chromosomal origins is transported through the stalk (followed by the rest of the chromosome) and positioned at the stalk-distal pole of the bud compartment. Of particular interest, this study nicely demonstrates that both steps initiate at or near the origin.

The manuscript is extremely thorough and covers a lot of ground, documenting the localization dynamics of ParB, ParA, PopZ, and chromosomal loci. The work also investigates the dynamics of DNA replication. Although the results are principally descriptive I think they provide new information that is unique to this unusual bacterium (or possibly bacteria in general that employ this mode of growth). Furthermore, the work raises as many new and interesting questions as it answers and lays the groundwork for more mechanistic studies in the future.

The writing is very clear and easy to follow - appropriate for a broad audience. Further, the quality of the data, especially the cytological analysis, is quite high and made it a pleasure to read (despite the presentation of so much data).

Below are some more minor concerns that I encourage the authors to address:

1) I would include topoisomerases as key players in chromosome segregation (with their ability to resolve rather than tangle DNA likely driven by the juxtaposition of chromosome "arms" by SMC complexes).

We have now added DNA topoisomerases to the list of important factors contributing to bulk chromosome segregation in the Introduction.

2) Does *H. neptunium* have an SMC complex? Might be worth mentioning/discussing.

H. neptunium has an SMC homolog (HNE_1917), which appears to be essential (unpublished). We now mention this fact in the Discussion.

3) I found the analogy to eukaryotic mitosis ("reminiscent of eukaryotic mitosis" was mentioned in the Abstract, Intro, Results, and Discussion) not particularly compelling. While it is true that the 2nd phase of segregation occurs after S-phase is mostly complete, the chromosomes are not cohesed at this stage as is the case in eukaryotes. Accordingly, the only similarity is the timing of segregation relative to replication. To my mind this process much more closely resembles spore formation in *B. subtilis* in which one of the replicated chromosome (or at least most of it) is transferred into the spore compartment after replication is complete during a late stage in cytokinesis.

We agree that origin segregation in *H. neptunium* is not directly comparable to eukaryotic mitosis because sister loci do not cohere before segregation. However, similar to mitosis, the two sister chromosomes remain closely associated after replication and their segregation involves an active process in which one of the two copies is, in its entirety, moved to a different (and distant) subcellular region. During spore formation in *B. subtilis*, by contrast, the origin region is already prepositioned at the old cell pole. The bulk of chromosomal DNA then follows across the closing forespore septum, in a process similar to FtsK-mediated DNA segregation across the regular division septum in other species. Overall, we still think that chromosome segregation in *H. neptunium* does indeed share some of the characteristics of eukaryotic mitosis. We would therefore like to retain references to this fact in the abstract and the first paragraph of the Discussion. However, we have now removed the expression “reminiscent of eukaryotic mitosis” from other parts of the paper.

4) The observation that one of the fluorescent ParB proteins (mCherry-ParBpMT1) used for FROS was stripped off the chromosome is reminiscent of the stripping function of FtsK observed in *B. subtilis* sporulation. I was surprised that the authors did not discuss a possible role for this DNA translocase in transporting the DNA through the stalk in the Discussion (I assume and FtsK is present in this bacterium). Especially because they show that a FROS signal associated with the *ter* region is consistently lost when it enters the stalk - suggesting that FtsK/XerCD dimer resolution occurs at the stalk base. Thus, FtsK would be well positioned to promote (or even drive) DNA transfer.

We thank reviewer #1 for pointing out this previous finding, which nicely shows that the sequential arrival of genes in an isolated compartment can establish a defined temporal pattern of gene expression with relevant physiological consequences. We have now included a description of the previous findings in the Discussion.

5) The observation that origin DNA enters the bud prior to the rest of the chromosome prompted the authors to speculate that this manner of chromosome segregation could establish a fixed temporal pattern of gene expression that helps to control bud development. This would be a good place to mention the transient genetic asymmetry observed in *B. subtilis* during spore formation. The original description of this phenomenon comes from Patrick Stragier (Frandsen et al 1999) but the demonstration of its biological relevance comes from Patrick Piggot and A. Hofmeister (Zupancic 2001, Khvorova et al 2000). I like the idea that *H. neptunium* might similarly take advantage of transient genetic asymmetry - or at least place specific genes closer to the origin (than *C. crescentus* does) to ensure sufficient quantities are produced as soon as possible.

Thank you for pointing out the previous work performed in *B. subtilis*. We now refer to these studies in the Discussion.

Reviewer #2 (Remarks to the Author):

General comment

In their manuscript Jung et al. combine bioinformatics analysis, biochemical approaches and microscopy to unveil the molecular actors and mechanisms regulating chromosome segregation in the *Hyphomonas neptunium* bacterium. Interestingly they report that in *H. neptunium* the

segregation of the replicated chromosome occurs in a two-step manner. This is completely novel in bacteria and opens the door to new perspectives in the study of bacterial chromosomal dynamics.

In some portions of the manuscript the authors emphasize that there is an uncoupling between DNA synthesis and chromosome segregation and I believe that this can not be unambiguously concluded from their results. Although the hypothesis remains reasonable from the data, I am more inclined to interpret the results as a partial uncoupling.

We agree that the replication and final segregation of chromosomal DNA are not completely uncoupled, because in many cases replication still continues for a short while after the start of the second segregation step. To make this clearer, we now state that the two processes are “largely” uncoupled.

Overall, this study is very well executed, thoroughly controlled, and the conclusions drawn are sound.

I point below some questions/comments/suggestions for the authors hoping it will help to clarify some aspects of the manuscript.

Data analysis:

In all figures where the authors perform a demographic analysis of the localization of protein complexes the color coded scale is confusing. What does intensity means in those scales? Fluorescence intensity?

Yes, the color code indicates the normalized fluorescence intensity. We have now modified the figures and replaced the label “intensity” with “signal intensity”.

Why did the authors choose not to find the center of mass for each foci (for ParB and PopZ)? This will give richer spatial information.

PopZ disperses throughout the cell before its relocation to the bud compartment and is thus difficult to localize by determination of the center of mass at some stages of the cell cycle. Apart from that, due to the unusual cell shape of *H. neptunium*, it was not possible to use standard software such as Oufi or MicrobeJ to analyze the subcellular positions of foci, necessitating manual analysis. It was not straightforward to determine the centers of mass manually with high precision. Apart from that, plotting the data is difficult because the number of foci varies over the course of the cell cycle (e.g. Figure 1C). Moreover, the centers of mass are often not perfectly clear in cells containing multiple foci (e.g. Figure 5D). Therefore, we chose to use line scans and then represent the data in demographs to provide a global view of the distribution of fluorescence in the different cell types. Although the resolution is lower in this case, it is still high enough to support the conclusions drawn on the basis of individual snap-shots or time lapse series.

In figure 7 it is unclear if the same demographic strategy was employed to localize the position of chromosomal loci.

To normalize the subcellular positions of the tagged loci to cell length (Figure 7B), it was necessary to determine the centers of mass of the mCherry-ParB_{pMT1} signals. We have modified the legend to Figure 7 to clarify the approach we used.

In figure 8 more quantitative information could be obtained, and perhaps easier to understand for a wider audience, if a plot of distance vs. time for each studied pair of loci was included. This will also allow to compare the velocity of displacement of each chromosomal region, if the distance between pairs of loci remains constant through cell cycle and perhaps to gain additional mechanistic insight.

To draw any firm conclusions, such a quantitative analysis would have to be performed on multiple cells per strain. However, the high variability of the mother cell and stalk lengths makes it difficult to directly compare data from different cells. Moreover, most of the movements observed occur within a very short time frame and are poorly resolved in our time-lapse series (Figure 8). Due to the faintness of the signals and considerable bleaching during the imaging process, it was not possible to reduce the acquisition intervals in order to increase the temporal resolution. For these reasons, we would prefer to just show representative images and not give any quantitative measures.

In figure 2C panel 2 it appears as the cell displays two foci at the stalk. How do the authors explain this? Was this behavior present in a large majority of cells?

This transient splitting of the ParB-*parS* complex into two distinct foci is observed regularly at onset of the second segregation step (see also panel E, 3 min). The precise percentage of cells showing this behavior is unknown, because the segregation process is very fast, making it difficult to capture the cell precisely at the stage when the splitting occurs. The molecular basis of this phenomenon remains to be determined. It is conceivable that, as the chromosomal region associated with ParB enters the stalk, ParB is progressively stripped off the DNA by the (so-far hypothetical) translocation machinery at the stalk base but then re-associates with its binding sites in the stalk lumen or the nascent bud compartment, giving rise to a new, second ParB focus. As DNA translocation continues, the focus in the mother cell gradually disappears and all of the ParB molecules become incorporated into the new complex.

The authors report two *parS* sites for *H. neptunium*. Most bacteria and plasmids carrying a ParABS system contain several repeats of *parS*. Moreover, a recent study confirmed the existence of additional *parS* sites in *C. crescentus* (10.1093/nar/gkx1192). Did the authors explore the possibility of additional *parS* sites in *H. neptunium*?

We searched an 80 kb region surrounding the two perfect *parS* sites shown in Figure 1B for the *parS* core motif 5'-CACGTGAAA-3', which was contained in most of the minor *parS* sites recently identified in *C. crescentus* (Tran *et al*, 2018). This analysis yielded only one additional site ~ 700 bp downstream of *parS*₂, suggesting that minor *parS* sites are infrequent in *H. neptunium*. Since the functionality of this additional site remains unclear, we would prefer not to include it in Figure 1A. Experimental approaches to detect potential minor ParB sites are beyond the scope of the present study, because the detection of minor *parS* sites would not help to further our understanding of the processes described in the paper.

In figure 5 panel D it was unclear what the black color indicates. I am not sure if colocalization is the correct way to call this analysis. The authors are not really interested in colocalization as DAPI signal is faint and distributed in the whole cell.

Black color indicates the absence of specific signals. We have added this information to the legend to Figure 5D. We agree that “colocalization” is not the correct term. We have now changed

“Colocalization” to “Localization”.

Manuscript organization:

The introduction may be shortened and more focus on the central aim of the paper, particularly the long paragraph devoted to *C. crescentus*. If comparison between the chromosome segregation mechanisms of *C. crescentus* and *H. neptunium* is pertinent I suggest to develop that in the discussion section.

We agree that the presentation of origin segregation in *C. crescentus* in the Introduction was too lengthy and distracted from the main point of the paper. We have therefore shortened it considerably, while still retaining some information on the ParABS system, which is (in our eyes) crucial to make the paper understandable to a general audience.

Although I found all figures relevant I wonder if there could be a way to fusion some of them in order to make the article more compact and easy to follow for a wider audience.

Each of the figures covers a different aspect of the results. A combination of their content would render the paper less structured and more difficult to follow. Moreover, it would require the removal of multiple panels from the main text to the supplement. Since all of the information presented in the main figures is relevant (as pointed out in the comment above), the reader would have to go back and forth between main and supplementary figures. For the sake of clarity, and since reviewer #1 found the paper very clear and easy to follow, we would therefore prefer to leave the figures as they are now.

Discussion:

The authors indicate that ParA, as it has been reported previously, requires the DNA scaffold to ‘travel’ and create a gradient. The authors address this in the discussion, however I found puzzling that in figure S2 on the fourth, fifth and sixth lines of panels (from the right, cell cycle progression) there is ParA signal on the bud while the two origins remain in the mother cell. How do the authors explain this?

Work in *C. crescentus* has shown that ParABS-mediated origin segregation involves polar landmark proteins (TipN and PopZ) that sequester free ParA monomers to prevent the spontaneous formation of new ParA dimers in the wake of the moving ParB-*parS* complex, thereby maintaining the directionality of the segregation process. We assume that the ParA-Venus focus observed in the nascent bud of *H. neptunium* cells is generated by a similar mechanism, involving a so-far unknown factor that localizes to the stalk-distal bud pole at the onset of bud formation. This expansion has now been included in the description of the results shown in Supplementary figure 2.

The results of measurements of translocation speed in the stalk strongly argue for an active mechanism. In the discussion the authors suggest the existence of an unknown mechanism. Does *H. neptunium* possess any FtsK-like translocase gene that could be a candidate for such an active mechanism?

We have now included a short paragraph on the potential role of FtsK in the translocation of chromosomal DNA through the stalk in the Discussion.

Does *H. neptunium* have SMC-like candidates?

Yes. The *H. neptunium* chromosome contains an ORF encoding an SMC homolog (HNE_1917). Unpublished work from our laboratory suggests that this protein is essential, but its precise role is so far unclear. We have now added this information to the Discussion.

Minor points:

The legend of Figure 1 has inverted the order of the panels C and D.

The legend has been corrected.

In all microscopy figures. Could the authors add an inset in each of them showing a zoomed individual and representative bacteria to be able to appreciate the foci size and shape (maybe with a dark background as they do in some other figures)? In all cases the bacteria images are very small and it is hard to see additional details.

We chose to show larger fields of cells to provide a representative overview of the phenotypes observed, especially if the phenotypes were heterogeneous and/or depended on the cell cycle stage. In all cases, both the cell shape as reflected by the DIC images and the fluorescence signals are important for the conclusions drawn. The addition of insets that show all the distinct phenotypes/cell cycle stages represented in the images plus additional images for the fluorescence channel would make the figures even larger and busier. We would therefore prefer to leave out insets and rather refer the reader to the online version of the article which can be easily viewed at higher magnification.

Page 9, line 195. The figure called there should be 4E.

Thank you for pointing out this error. We now refer to the correct figure.

Could the authors include the full name of the microscopy technique the first time it appears mentioned in the text in legend of Figure 2 'Differential interference contrast (DIC) microscopy'?

Done.

The movie S1 was not present among the files received for the reviewing process.

We are sorry for that. Supplementary movie 1 is now included in the submission.

Reviewer #3 (Remarks to the Author):

The manuscript from the Thanbichler's lab interrogates a fundamental biological process that is surprising poorly understood in bacteria, which is the mechanism of chromosome segregation. Previous work by the Thanbichler lab revealed important insights into DNA segregation in the related alpha proteobacteria *Caulobacter crescentus*. In this manuscript, the authors did a large amount of work focusing on the alphaproteobacteria *H. neptunium*, for which tools have only recently been developed, enabling the cellular dissection of its biological processes. This cell biological study reveals that the *parABS* partition system of *H. neptunium* is essential for DNA segregation and that it employs a two step process for segregation involving bud formation. Given

how little we know about bacterial segregation, I believe this work represents an important contribution to the field. Therefore I am in favor of publication but have just a few questions to be addressed.

1. While the *H. neptunium* appears to have a PopZ, the analyzes suggests it is functionally different from the *c. crescentus* PopZ. But it was unclear to me from their studies whether it interacts with ParB as does the *Caulobacter* PopZ? do they see the protein colocalize?

To determine whether the two proteins are closely associated *in vivo*, we performed localization studies on a newly constructed strain (AJ90) producing both a ParB-Cerulean and a PopZ-Venus fusion. Our results show that PopZ perfectly colocalizes with the polar ParB-*parS* complex in swarmer (G1-phase) cells and, at a later stage, in the bud compartment. These results suggest that PopZ and ParB interact, directly or indirectly, with each other. The data are now shown in the new Supplementary Figure 5 and described in the Results section.

2. The authors say that the studies in Fig.7 show that the *H. neptunium* chromosome is circular. But it doesn't seem like these studies have the resolution to make that claim. Clearly the chromosome is not tightly condensed with Ori and Ter in the same volume for example, but the chromosome could still be compacted with a elongated, packed structure, based on the results.

We agree that it is not possible to draw any definitive conclusions on the precise arrangement of chromosomal DNA within the cell based on our localization data. To account for both possibilities, we now state that the chromosome "appears to be organized into a ring-like or elongated structure".

3. Also, the image at 186 degrees (Fig. 7) seems to be missing the *parS* signal. Is that because the *parS* region is overlapping with the region of the chromosome?

For unknown reasons, it was not possible to introduce the constructs required for labeling the chromosomal 186° position into cells carrying the *parB-yfp* fusion. In this case, the localization analysis was therefore performed with a strain (AJ49) that still carried the native *parB* gene, explaining the absence of the YFP signal. We now point out this fact more clearly in the legend to Figure 7C. It was still possible to unambiguously identify the old and new cell poles and thus properly map the mCherry-ParB_{pMT1} signal (in Figure 7B) in the wild-type background based on the fact that the new pole of the swarmer cell is consistently more tapered than the old pole.

4. On page 6 the authors say they generated a strain producing a fully functional ParB-YFP fusion in place of the native ParB. But what is the data showing that? Maybe I missed that.

In the strain used in this study, the native *parB* gene was replaced with a *parB-eyfp* fusion. Given that this strain still grows normally, we assume that the fluorescence tag does not impair the functionality of the protein. However, we agree that detailed *in vitro* analyses would be required to verify that the properties of the fusion protein are indeed identical to that of the wild-type protein. Therefore, we have now removed the word "fully" and only state that the protein is "functional".

5. In figure 1 the legend for C should be D and vice versa; the legends are reversed.

Thank you for point out this inconsistency. We have corrected the legend as requested.

REVIEWERS' COMMENTS:

Reviewer #1 (Remarks to the Author):

The authors have addressed all of my concerns in a satisfactory manner. I look forward to seeing this thorough and interesting paper in "print".

Reviewer #2 (Remarks to the Author):

The authors have addressed all my comments and remarks regarding the key elements of the manuscript.

As a minor detail, the authors answered one question regarding the role of ParA and a similar mechanism in *C. crescentus* and indicate that this discussion is now included in Supplementary figure 2. However, I did not find this new content in the current version of the supplementary material. Could the authors please correct this before the final submission?

Reviewer #3 (Remarks to the Author):

The revised manuscript from Jung and coworkers, I believe, has addressed the reviewer's questions and any concerns. The work is very solid and will be of interest to a wide audience. The work represents an important contribution to the field.

REVIEWERS' COMMENTS:

Reviewer #1 (Remarks to the Author):

The authors have addressed all of my concerns in a satisfactory manner. I look forward to seeing this thorough and interesting paper in "print".

Thank you for your favorable comments.

Reviewer #2 (Remarks to the Author):

The authors have addressed all my comments and remarks regarding the key elements of the manuscript.

As a minor detail, the authors answered one question regarding the role of ParA and a similar mechanism in *C. crescentus* and indicate that this discussion is now included in Supplementary figure 2. However, I did not find this new content in the current version of the supplementary material. Could the authors please correct this before the final submission?

Thank you for your favorable comments. We are sorry if our response was not clear. We did not add an additional Supplementary figure but included a short discussion of the results shown in Supplementary Figure 2 in the Results section.

Reviewer #3 (Remarks to the Author):

The revised manuscript from Jung and coworkers, I believe, has addressed the reviewer's questions and any concerns. The work is very solid and will be of interest to a wide audience. The work represents an important contribution to the field.

Thank you for your favorable comments.